# REASONING-INTENSIVE REGRESSION

## ABSTRACT

AI researchers and practitioners increasingly apply large language models (LLMs) to what we call reasoning-intensive regression (RiR), i.e., deducing subtle numerical scores from text. Unlike standard language regression tasks, e.g., for sentiment or similarity, RiR often appears instead in ad-hoc problems such as rubric-based scoring, modeling dense rewards in complex environments, or domain-specific retrieval, where much deeper analysis of context is required while only limited task-specific training data and computation are available. We cast four realistic problems as RiR tasks to establish an initial benchmark, and use that to test our hypothesis that prompting frozen LLMs and finetuning Transformer encoders via gradient descent will both often struggle in RiR. We then propose MENTAT, a simple and lightweight method that combines batch-reflective prompt optimization with neural ensemble learning. MENTAT achieves up to $65\%$ improvement over both baselines, though substantial room remains for future advances in RiR.

## 1 INTRODUCTION

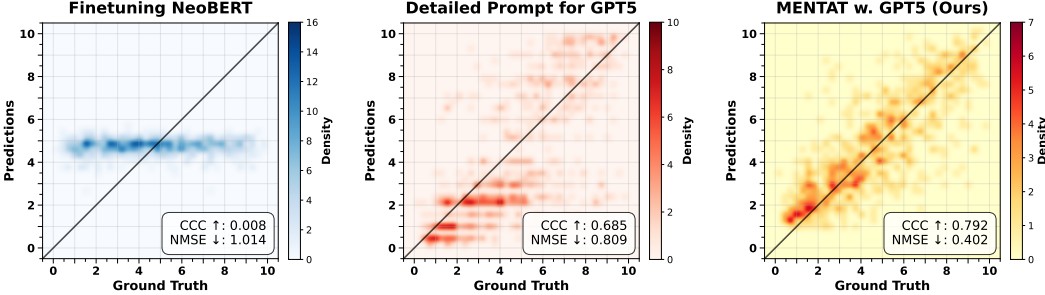

Figure 1: On regression for detecting the first math error, finetuning a NeoBERT model collapses to mean predictions (CCC = 0.01). Meanwhile, detailed (human-crafted) prompting achieves reasonable concordance (CCC = 0.69) but exhibits coarse and imprecise prediction behavior (the dense horizontal lines and near-random NMSE). MENTAT's performance illustrates how RiR problems benefit from combining deep reasoning capabilities with precise numerical predictions.

Despite fast progress in adapting large language models (LLMs) for building downstream AI systems, lightweight methods for adapting LLMs to even standard *natural-language regression* tasks remain surprisingly elusive (Lukasik et al., 2024b;a; Tang et al., 2024; Song et al., 2025; Song & Bahri, 2025). These tasks, like sentiment analysis, semantic similarity, and document ranking, involve predicting a score $y \in \mathbb{R}$ from a natural-language string. Surprisingly, on these problems, applying straightforward supervised learning to pretrained Transformer encoders such as BERT Devlin et al. (2019) has been shown to perform competitively with much larger decoder-only LLMs Lukasik et al. (2024a), even with sophisticated fine-tuning methods.

We investigate what we call *Reasoning-Intensive Regression* (RiR), a fuzzy but growing subset of natural-language regression in which processing the text in each instance *demands sequential deduction or deep analysis, rather than shallow identification of features*. Unlike simpler regression tasks, RiR problems call for explicit step-by-step problem decomposition or *reasoning*, where the system produces intermediate sequences of steps like tokens $\langle r_1, ..., r_t \rangle \in \Sigma^*$ before committing to a

prediction (Merrill & Sabharwal, 2024). See Figure 2 for a breakdown of regression problems into three levels of complexity: feature-based, semantic analysis, and reasoning-intensive, inspired by Su et al. (2025)'s analysis of retrieval tasks.

These types of applications are emerging rapidly in both research and practice, e.g. to produce scores for ad-hoc applications that process customer calls, student essays, rubric-based LLM generation, or instruction-based query–document relevance (MacDonald, 2024; Es et al., 2024; Su et al., 2025; Thakur et al., 2025). In parallel, the same scoring paradigm is being scaled in recent efforts toward general-purpose chain-of-thought reward models Kimi Team (2025); Ankner et al. (2024a), but these typically assume orders-of-magnitude more labels and compute (e.g., hundreds of thousands of labels in K2) than the lightweight application-specific regimes that are far more common in the long tail.

We establish an initial benchmark for RiR by casting four realistic tasks as regression problems that demand *varying* levels of reasoning: predicting the proportion of a long mathematical deduction up to the first erroneous statement, determining the extent to which an LLM can follow highly composite instructions, predicting the degree to which the response of one Retrieval-Augmented Generation (RAG) system is better than another, and grading student essays on supplied topics. We then identify two practical constraints of downstream applications of RiR: these applications tend to offer only (very) small training sets and have room only for accessible and lightweight computations like LLM inference, lightweight forms of LLM prompt optimization, and finetuning medium-sized neural networks such as small Transformers, but not, say, large-scale reinforcement learning for large language models (DeepSeek-AI, 2025; Kimi Team, 2025).

We ask: *Are there effective methods that are data- and compute-efficient for tackling ad-hoc reasoning-intensive regression problems?* We hypothesize that what makes RiR problems especially challenging is that they combine the reasoning need for *deep analysis of each individual task instance* with the regression challenge of *learning to produce precise, calibrated, and well-ranked scores* from very little data. As illustrated in Figure 1, standard prompt engineering techniques struggle with the high precision needed for learning to approximate a statistical distribution, while approaches that bypass LLM-based reasoning, e.g., training small Transformer encoders, often fail to truly learn RiR problems and instead seek to "hack" the regression loss function by finding degenerate approximations (e.g., collapsing to a small range of scores).

We propose **M**istake-**A**ware prompt **E**volver with **N**eural **T**raining **A**nd **T**esting (MENTAT), a simple and lightweight method that combines iterative prompt optimization with neural regression. Rather than relying on LLMs to produce precise numerical predictions directly, which often results in brittle outputs, MENTAT uses an iterative error-driven prompt evolution process. Starting with even just a very basic prompt, the LLM analyzes its own prediction errors *in large batches*, identifies patterns of its poor performance, and then refines the prompt based on that. After few iterations, MENTAT trains a simple aggregation MLP to reduce multiple rollouts from the LLM-discovered prompt into a final prediction. MENTAT delivers consistent improvements in quality, but nonetheless leaves large headroom on many of the RiR settings we define.

The remainder of the study is as follows: Section 2 describes how we translate four problems into RiR tasks and Section 3 introduces MENTAT. Section 4 presents our evaluation methodology, including the details of our baselines, and the results. The paper concludes with Sections 5 and 6, which synthesize our findings and discuss implications for future research. An extended discussion of related work is given in the Appendix A.

## 2 BENCHMARKING RiR

We collect four tasks for Reasoning-Intensive Regression of varying degrees of reasoning intensity. Refer to Figure 3 for the dataset distributions.

- **Mathematical Error Detection** requires precise logical reasoning and stepwise analysis, while also stressing the fact that LLMs are known to struggle with precisely estimating simple properties like text length.

- **Instruction Following** evaluates how well a response satisfies a set of fine-grained requirements, and expects models to produce calibrated scalar judgments.

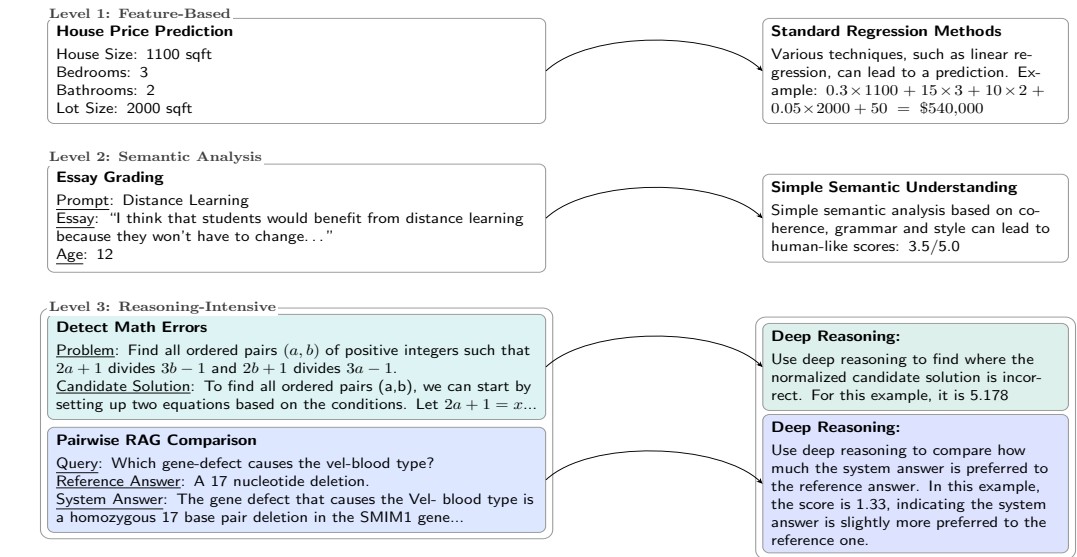

Figure 2: Inspired by Su et al. (2025)'s analysis of retrieval tasks, we break down text-based regression problems into three, informal complexity levels. Level 1 tasks use simple feature-based inputs (for example, the number of bedrooms and bathrooms when predicting home prices). Text-to-text regression achieves strong Level 1 performance with rich datasets (Akhauri et al., 2025). Level 2 tasks require moderate semantic understanding (sentiment analysis, reward modeling) but are easy for supervised-learning over a pretrained Transformer. Level 3, the focus of this work, represents Reasoning-Intensive Regression (RiR), which requires deep sequential reasoning.

- **Pairwise RAG Comparison** emphasizes the need for models to perform nuanced human-like judgment and contextual understanding, demanding sophisticated evaluative reasoning.

- **Essay Grading** serves as a reference point, requiring semantic understanding where encoders like BERT might already perform well with a reasonable amount of finetuning data.

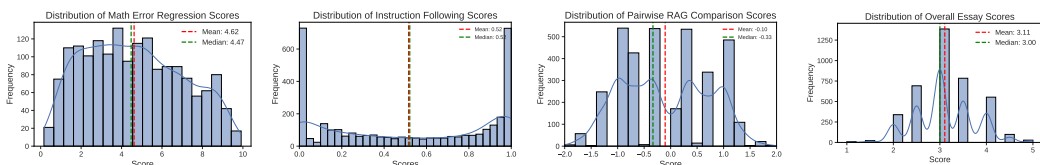

Figure 3: Ground-truth score distributions for mathematical error detection (the spread capturing the tendency for solutions to fail towards the center), instruction following (capturing the tendency to favor the tails), pairwise RAG comparison (narrow distribution around averaged judgments), and essay grading (tight clustering characteristic of qualitative assessments).

**Regression Metrics** Normalized Mean Square Error (NMSE) is a common metric for reporting regression performance: $\sum_i^n (y_i - \hat{y}_i)^2 / \sum_i^n (y_i - \bar{y})^2$, where $n$ is the size of the dataset, $\hat{y}_i$ is a prediction, $y_i$ the corresponding ground truth value, and $\bar{y}$ is the mean of the set.

But distance-based metrics are inadequate for typical RiR problems; RiR systems can artificially lower their NMSE simply by avoiding "risky" predictions at the extremes. This can be seen in Figure 1 earlier, particularly in comparing the fine-tuned NeoBERT model Breton et al. (2025) against detailed (human-crafted) prompting. Following Figure 1, if we were to rely on NMSE, detailed prompting for GPT-5 would not appear to substantially outperform NeoBERT (0.81 vs. 1.01), and this gap would be even reversed for weaker LLMs. Examining the distribution of predictions reveals

that NeoBERT "hacked" the loss function by learning a collapsed distribution, while the prompted LLM actually shows substantial signs of ranking the inputs correctly.

This can be captured in a Concordance Correlation Coefficient (CCC) of $0.01$ for NeoBERT versus a CCC of $0.69$ for detailed prompting. We thus suggest the use of the CCC as an additional, and perhaps more appropriate, RiR metric. CCC measures both correlation and agreement, defined as $\frac{2\rho\sigma_y\sigma_{\hat{y}}}{\sigma_y^2+\sigma_{\hat{y}}^2+(\mu_y-\mu_{\hat{y}})^2}$, where $\rho$ is the Pearson correlation coefficient between predictions $\hat{y}$ and ground truth $y$, $\sigma_y$ and $\sigma_{\hat{y}}$ are their respective standard deviations, and $\mu_y$ and $\mu_{\hat{y}}$ are their means. CCC penalizes systematic bias and rewards predictions that maintain the natural variance of the distribution.

**Detecting Mathematical Errors** We derive a dataset for predicting the *fraction* of a mathematical solution up to the first erroneous reasoning step, given a problem and incorrect solution in LaTeX, from ProcessBench (Zheng et al., 2024). To effectively do this, a model must systematically reason formally about math steps rather than relying on probabilistic heuristics, but it must also be good at estimating relative lengths and inferring the boundaries of the steps in a calibrated way.

To convert the original classification task into a regression problem, we first filter out problems with correct solutions or final answers. We then merge all solution steps into a single continuous text $T = s_1\|s_2\|\cdots\|s_n$ (here $\|$ denotes concatenation). Next, for a solution with error at step $k$, the regression score $R$ is $10 \times \left(\sum_{i=1}^{k-1}|s_1| + \frac{1}{2}|s_k|\right)/|T|$ where $|s_1|$ denotes the length of step $i$, and $|T|$ is the total length of the concatenated solution. See an example entry in Appendix E.

**Instruction Following** We derive a task from the WILDIFEVA corpus Lior et al. (2025) that targets *instruction-following* in long-form generation. Each example consists of: (i) a user *task* prompt; (ii) a list of atomic requirements (the *decomposition*); (iii) a *model answer* produced by Llama-3.1-8B (zero-shot); and (iv) per-requirement satisfaction scores originally produced by Llama-3.1-70B acting as an automatic judge. The goal is to predict a single continuous label $y \in [0,1]$ that reflects the overall degree to which the answer adheres to the decomposed instructions. More precisely, for each decomposition instance, the judge produced a probability-like score $s_i \in [0,1]$ for each requirement $r_i$, $i = 1, \ldots, K$. We then use the harmonic mean of these scores as our overall judgment, emphasizing the need to adhere well to all task requirements. To test instruction following, we do not expose the decomposition to NeoBERT or an LLM; instead, they are only given the task and model answer and must infer the overall score.

**Pairwise RAG Comparison** We derive a dataset for comparing two LLM outputs on a scale from the RAG-QA evaluations (Han et al., 2024). Each query $q \in \mathcal{Q}$ has responses $A_1, A_2$ and a target comparative score from $-2$ to $2$ representing the *average* annotation of three human judges, who were instructed to assess response helpfulness, truthfulness, and completeness. Here, positive scores means that $A_1$ is better (and vice versa). This task partially aligns with RiR as judging the outputs and comparing them in light of each query often requires nuanced judgment.

**Essay Grading** We lastly use an essay grading dataset Crossley et al. (2023), where each entry contains among other features an essay prompt, a student (grade 8–12) response, associated demographic information, and an overall score between 1 and 5. Although Essay Grading is simpler than the rest, it serves as a reference point for the other RiR tasks.

We evaluate these tasks using two proprietary LLMs (`GPT-4.1`, `GPT-5`) across three tasks, plus an open-source model (`gpt-oss-20b`) on Instruction Following for reproducibility and generalization validation.

## 3 MENTAT

MENTAT combines two simple ideas, depicted in Figure 4: it allows the LLM itself to reflect in batches to incrementally adjust its own prompt, and it aggregates multiple rollouts from the optimized LLM system with a simple trained MLP.

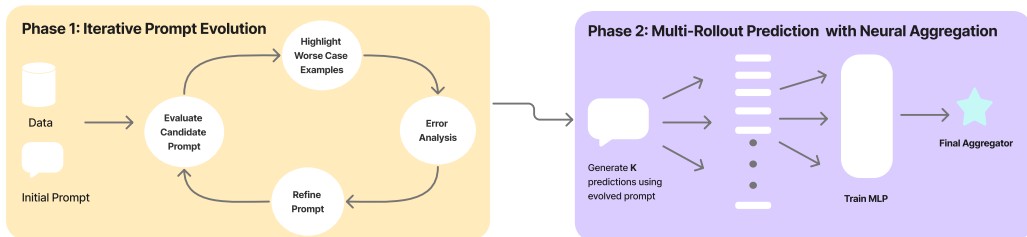

Figure 4: Phase 1 performs prompt evolution through iterative and *batched* reflection. Given a candidate prompt, we collect rollouts on $n$ samples, divided into training and validation sets. A model instructed to focus on the $\sqrt{n}$ worst-performing examples (selected based on absolute prediction error) analyzes the rollouts on the training samples, in light of the optimization history from previous iterations, and makes proposals that refine the prompt. This cycle continues for a predetermined number of iterations, after which we select the best-performing prompt $P_{\text{best}}$ as evaluated on the validation set , where by best we mean the prompt that led to the highest CCC value. Phase 2 generates multi-rollout predictions by applying $P_{\text{best}}$ and aggregating $K$ stochastic predictions per input and trains a neural aggregator $f_\theta$ on sorted rollouts using a combined CCC–NMSE loss. Test predictions are obtained by sampling test rollouts and applying the trained aggregator $f_{\theta^\star}$.

**Phase 1: Prompt Evolution**  MENTAT's first step is to make sure that the LLM prompt reflects both *local* instructions for reasoning about each input and *global* guidance about the distribution of ground-truth scores. Though any approach for prompt optimization can be used here, e.g., MIPRO Opsahl-Ong et al. (2024) or GEPA Agrawal et al. (2025), through preliminary experiments we identified two special properties in RiR tasks that call for different design choices.

First, performing rollouts with powerful *reasoning* models can be expensive and slow, when compared to standard LLMs, for which existing optimizers were built. To remain within the lightweight constraints of typical RiR tasks, a suitable prompt evolution stage would have to minimize *both* the number of rollouts performed with the LLM and the number of *inherently sequential stages* or iterations of optimization. Second, RiR tasks require attention to distributional properties, calibration, variance matching, and avoiding collapse to mean predictions, beyond per-example accuracy. This is because MENTAT's aggregation design demonstrates that it can be easy to turn a *well-calibrated* system into one that has low pointwise error, but the reverse is not necessarily true.

This motivates us to test an exceedingly simple reasoning-based technique for optimizing LLM systems that contain a single prompt.[1] While batch-based prompt optimization has been extensively explored in prior work Pryzant et al. (2023); Ye et al. (2024), we focus on combining it with neural aggregation specifically for regression tasks, using CCC alongside NMSE to guide prompt selection and aggregator training. This simple design is inspired by human prompt engineering practice (Husain & Shankar, 2024).

Concretely, we proceed in a very small number of sequential iterations (three in our experiments). In each iteration, the work is highly parallelizable: we evaluate the current prompt on a shuffled sample of the training set, and then concatenate all of the rollouts for analysis by the same LLM. It is then asked to identify systematic errors by analyzing the worst-performing examples and to generate improved instructions. In each iteration, the LLM receives three key inputs: current instructions, performance analysis with detailed error patterns, and a formatted history of previous optimization attempts. This historical context prevents the method from cycling through previously unsuccessful approaches and enables progressive refinement. At the end of this process, the best-performing prompt (via NMSE or CCC) is selected on a separate validation set.

In our evaluation, to stress MENTAT, we start from a deliberately basic prompt for each task, to reflect a more challenging setting.[2] Note also that this iterative prompt evolution follows a single optimization

---

[1]We leave extending this method to multi-stage LLM programs and conducting an extensive comparison of different prompt optimization strategies to future work.

[2]Examples of the basic vs. the detailed prompts used for the four tasks can be found in Appendices H and F), respectively. They differ in the inclusion of detailed procedural steps, calibration guidance, and/or domain-specific heuristics that human experts may decide to include.

trajectory. In principle, MENTAT could employ multiple random restarts, which could be parallelized to explore diverse regions of the prompt space. However, we focus on single-trajectory optimization both for computational efficiency and algorithmic simplicity.

**Phase 2: Multi-Rollout Generation with Neural Aggregation**  Using the best LLM-discovered prompt from Phase 1, MENTAT generates multiple independent predictions for each example. The multi-rollout approach captures the inherent uncertainty in LLM predictions, as each rollout can reason independently, and provides richer signal for the subsequent neural aggregation phase. In practice, we set this to three rollouts per example.

We train a small Multi-Layer Perceptron (MLP) to combine rollout predictions. The aggregator ensures order invariance by sorting rollout predictions, incorporates statistical features (mean, standard deviation, min, max), and is optimized for a combination of the CCC and NMSE loss functions. Overall, this method builds on self-consistency Wang et al. (2023) and best-of-N voting Stiennon et al. (2020); Snell et al. (2024), but differs by training a lightweight aggregator that learns task-specific weighting of rollout statistics rather than using fixed aggregation rules.

**Analysis of MENTAT**  Figure 5 shows rollout variance distributions for detailed (human-crafted) versus MENTAT-evolved prompts across three tasks. MENTAT's prompt evolution consistently reduces variance on reasoning-intensive tasks, achieving a 30% reduction in mean variance on Mathematical Error Detection. This demonstrates that evolved prompts produce more stable reasoning patterns rather than merely providing noisy signals for the aggregator to smooth. However, non-trivial variance remains after evolution, enabling the neural aggregator to extract meaningful signal from rollout diversity. These findings reveal MENTAT's complementary design: prompt evolution improves prediction reliability while neural aggregation refines these consistent signals into precise numerical outputs.

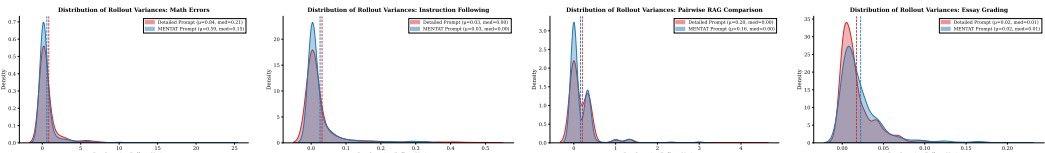

Figure 5: Distribution of per-question rollout variances comparing the Detailed (human-crafted) prompt against the MENTAT-evolved prompt across three tasks. For reasoning-intensive tasks (Mathematical Error Detection and Pairwise RAG Comparison), MENTAT's prompt evolution yields lower mean rollout variance, indicating more consistent predictions across independent rollouts. In contrast, Essay Grading, which is characterized as a Level 2 (semantic analysis) task requiring less sequential reasoning, shows comparable variance between prompts. This pattern suggests that prompt evolution yields the greatest consistency gains on tasks where deep reasoning is essential, while contributing less when shallow semantic features suffice.

## 4 EVALUATION

We define two standard baselines for RiR problems: fine-tuning a small Transformer encoder and prompting an LLM, and use these two to understand the relative merits of our method MENTAT and to develop a series of ablation experiments. Additionally, we compare against Agrawal et al. (2025), a recent reflective prompt optimization method, to situate MENTAT relative to modern prompt optimizers.

### 4.1 BASELINE: FINETUNING A TRANSFORMER ENCODER

We formulate RiR as supervised regression using a 250M-parameter NeoBERT model. The architecture processes minimally formatted text sequences (e.g., combining problem statements with solutions for math errors, augmented with domain-specific prompts).

Inputs are tokenized using NeoBERT's byte-level BPE tokenizer, truncated or padded to 1024 tokens, and passed through the pretrained encoder. The model extracts representations from the [CLS] token, applies dropout regularization ($p = 0.2$), and uses linear projection for scalar predictions.

The optimization objective minimizes weighted NMSE and CCC using AdamW (Loshchilov & Hutter, 2019). This architecture requires only prompt templating beyond standard fine-tuning, with hyperparameters detailed in Appendix D.1.

## 4.2 BASELINE: PROMPTING A LARGE LANGUAGE MODEL

We employ Chain-of-Thought style prompting to encourage frozen LLMs to perform explicit reasoning through step-by-step token generation. Our evaluation uses two proprietary models with different reasoning capabilities: GPT-4.1 (non-reasoning) and GPT-5 (reasoning) across three tasks (Mathematical Error Detection, Pairwise RAG Comparison, and Essay Grading). For the Instruction Following task, we employ gpt-oss-20b, an open-source model, to demonstrate that our method generalizes beyond proprietary systems and to provide more easily reproducible baselines for the community. The detailed prompts for all tasks help guide the decomposition of complex inputs and the templates can be found in Appendix F.

This approach is motivated by several practical advantages. Frozen LLMs can act as a unified interface across various natural language tasks, with no or very little training data. This is especially valuable in reasoning-intensive regression (RiR) tasks where annotated datasets are often scarce. Utilizing a shared, unified, and amortized infrastructure (i.e., LLM servers) enables us to deploy a single model across many tasks, significantly reducing the computational and financial overhead compared to training multiple specifiable models.

## 4.3 EXPERIMENTAL SETUP

Our experimental design evaluates MENTAT across four reasoning-intensive regression tasks using a structured approach. We test GPT-4.1 and GPT-5 on three tasks (Mathematical Error Detection, Pairwise RAG Comparison, and Essay Grading), while Instruction Following uses gpt-oss-20b to demonstrate generalization to open-source models.

For the three tasks using proprietary models, we employ 750 test examples with results averaged across three independent runs. We evaluate under two training configurations (100 and 500 samples) that reflect real-world data constraints typical in ad-hoc RiR applications. For prompt optimization methods (including MENTAT), we use balanced train/validation splits of $50+50$ and $250+250$ samples; Phase 1 uses these for prompt evolution, and Phase 2 generates 3 rollouts per training sample for MLP training. For NeoBERT finetuning, we employ training-heavy splits of $50 + 50$ and $350 + 150$ samples to leverage the model's supervised learning capabilities.

For Instruction Following, we use a single configuration with 500 training, 500 validation, and 2000 test samples, reflecting the different data availability typical for this fundamental capability assessment. This larger test set enables more robust evaluation of the nuanced instruction-adherence requirements.

This experimental structure allows us to assess MENTAT's effectiveness across different model capabilities (reasoning vs. non-reasoning), data regimes (limited vs. moderate training data), and model accessibility (proprietary vs. open-source), providing comprehensive validation of our approach for practical RiR applications.

### 4.3.1 COMPUTATION COST

MENTAT's computational costs comprise two phases. At inference time, each prediction requires 3 rollouts, resulting in $3\times$ token cost compared to single-pass prompting. However, all rollouts can be generated in parallel, so wall-clock latency remains approximately equivalent to a single rollout in parallelized deployment scenarios.

During optimization (Phase 1), MENTAT uses a fixed 3-iteration design. Each iteration evaluates the current prompt on all $n = 250$ samples (parallelizable) and performs one reflection call analyzing the $\sqrt{250} \approx 16$ worst-performing examples, totaling approximately 753 LLM calls across 3 sequential steps. GEPA's "light" configuration converges after an average of 23 sequential iterations (ranging 15-34 across runs). GEPA's evolutionary search thus requires approximately $8\times$ more sequential rounds than MENTAT's fixed design, providing MENTAT a substantial wall-clock advantage in parallelized deployments, though total token consumption may differ.

| LM | Method | Math Errors | | | | Pairwise RAG | | | | Essay Grading | | | |
|----|--------|-------------|--|--|--|--------------|--|--|--|---------------|--|--|--|
| | | NMSE ↓ | | CCC ↑ | | NMSE ↓ | | CCC ↑ | | NMSE ↓ | | CCC ↑ | |
| | | 100 | 500 | 100 | 500 | 100 | 500 | 100 | 500 | 100 | 500 | 100 | 500 |
| **Main Methods** | | | | | | | | | | | | | |
| NeoBERT | Gradient Descent | 1.05 | 1.01 | 0.02 | 0.06 | 1.44 | 1.02 | 0.02 | 0.10 | 1.03 | 0.91 | 0.19 | 0.65 |
| GPT-4.1 | Basic Prompt | 1.59 | 1.59 | 0.36 | 0.36 | 2.18 | 2.18 | 0.47 | 0.47 | 0.75 | 0.75 | 0.63 | 0.63 |
| | Detailed Prompt | 1.13 | 1.13 | 0.52 | 0.52 | 2.20 | 2.20 | 0.47 | 0.47 | 0.73 | 0.73 | 0.65 | 0.65 |
| | MENTAT$_{\text{Basic Prompt}}$ | 0.87 | 0.76 | 0.51 | 0.49 | **0.77** | **0.80** | **0.50** | **0.52** | **0.54** | **0.53** | **0.70** | **0.68** |
| GPT-5 | Basic Prompt | 0.77 | 0.77 | 0.66 | 0.66 | 2.25 | 2.25 | 0.35 | 0.35 | 1.31 | 1.31 | 0.42 | 0.42 |
| | Detailed Prompt | 0.78 | 0.78 | 0.69 | 0.69 | 2.18 | 2.18 | 0.31 | 0.31 | 1.53 | 1.53 | 0.40 | 0.40 |
| | MENTAT$_{\text{Basic Prompt}}$ | **0.52** | **0.42** | **0.72** | **0.78** | 1.07 | 0.93 | 0.36 | 0.33 | 0.64 | 0.67 | 0.59 | 0.55 |
| **Ablations** | | | | | | | | | | | | | |
| GPT-4.1 | MENTAT Prompt | 1.39 | 1.29 | 0.45 | 0.48 | 2.00 | 1.69 | 0.45 | 0.48 | 0.61 | 0.71 | 0.68 | 0.66 |
| | MENTAT-Avg | 1.00 | 1.01 | 0.52 | 0.52 | 1.82 | 1.48 | 0.48 | 0.51 | 0.57 | 0.63 | 0.69 | **0.68** |
| | GEPA | 1.04 | 1.01 | 0.49 | 0.54 | 2.16 | 2.40 | 0.44 | 0.43 | 0.79 | 0.81 | 0.63 | 0.63 |
| GPT-5 | MENTAT Prompt | 0.66 | 0.58 | 0.66 | 0.72 | 1.43 | 1.95 | 0.33 | 0.30 | 0.74 | 0.70 | 0.57 | 0.54 |
| | MENTAT-Avg | 0.59 | 0.51 | 0.68 | 0.75 | 1.31 | 1.83 | 0.35 | 0.32 | 0.69 | 0.67 | 0.57 | 0.55 |
| | GEPA | 0.78 | 0.63 | 0.68 | 0.69 | 2.48 | 2.29 | 0.28 | 0.28 | 1.01 | 1.01 | 0.42 | 0.44 |

Table 1: Performance comparison across Mathematical Error Detection, Pairwise RAG Comparison, and Essay Grading using `GPT-4.1` and `GPT-5` as our models. Each entry is the average of three independent runs on a test set of size 750. Total training sizes are 100 and 500 (train/val combined). **Ablations:** MENTAT Prompt uses only error-driven prompt refinement on training data *starting from a basic prompt*; MENTAT-Avg shows performance when replacing the trained MLP with averaging. We remark here that NeoBERT obtains an average NMSE and CCC of 0.60 and 0.66 respectively, on a training regime of 1500 (1000 training + 500 validation) on Pairwise RAG Comparison. That is, NeoBERT needs much more data on this task to lead to good performance, but it can be achieved. This table along with additional reporting of standard deviation can be found in Appendix 3.

| LM | Method | Instruction Following | |
|----|--------|-----------------------|--|
| | | NMSE ↓ | CCC ↑ |
| NeoBERT | Gradient Descent | 1.08 (0.07) | 0.36 (0.04) |
| GPT-OSS-20B | Basic Prompt | 1.18 (0.00) | 0.32 (0.00) |
| | Detailed Prompt | 1.16 (0.00) | 0.33 (0.00) |
| | MENTAT$_{\text{Basic Prompt}}$ | 0.95 (0.09) | 0.42 (0.01) |
| | MENTAT$_{\text{Detailed Prompt}}$ | **0.90 (0.04)** | 0.43 (0.00) |
| **Ablations** | | | |
| GPT-OSS-20B | MENTAT$_{\text{Basic Prompt}}$ Prompt | 1.25 (0.05) | 0.35 (0.01) |
| | MENTAT$_{\text{Basic Prompt}}$-Avg | 1.06 (0.04) | 0.38 (0.02) |
| | MENTAT$_{\text{Detailed Prompt}}$ Prompt | 1.24 (0.13) | 0.36 (0.01) |
| | MENTAT$_{\text{Detailed Prompt}}$-Avg | 1.09 (0.06) | 0.39 (0.02) |
| | GEPA | 1.06 (0.02) | **0.46 (0.01)** |

Table 2: Performance on the Instruction Following task using the `gpt-oss-20b` model. Each entry is the average of three independent runs on a test set of size 2000. Total training configuration uses 500 training and 500 validation samples. Ablations: MENTAT Prompt uses only error-driven prompt refinement on training data; MENTAT-Avg shows performance when replacing the trained MLP with averaging. The subscripts, basic prompt and detailed prompt, are what we use an the initial prompt in the MENTAT framework. Moreover, values within the parenthesis represent standard deviations.

Phase 2 (MLP training) has negligible cost, as the MLP contains only 8 hidden units and trains on 750 rollout vectors (250 samples $\times 3$ rollouts each).

## 4.4 RESULTS

Our main evaluation results are reported in Table 1 and 2, demonstrating significant performance variations across methods and tasks. The results reveal distinct patterns in how different approaches handle reasoning-intensive regression problems, with MENTAT consistently outperforming baseline methods across most configurations. Beyond aggregate metrics, we analyze failure modes across methods: NeoBERT's distribution collapse (1), GPT-5's center-seeking behavior on pairwise RAG

(6), and systematic quantization patterns in LLM outputs (Appendix B). We provide additional per-task qualitative error analysis in Appendix B.

**Mathematical Error Detection Performance**   On this task, finetuning NeoBERT achieves near-zero CCC scores across both training configurations and effectively collapsing to mean predictions as shown in Figure 1. In contrast, LLM-based approaches demonstrated substantial reasoning capabilities. GPT-4.1 with detailed prompting achieved CCC scores of $0.52$ (100-sample training) and maintained this performance at 500 samples. However, MENTAT with GPT-4.1 showed only modest improvements, reaching CCC scores of $0.51$ (100 samples) and $0.49$ (500 samples), representing approximately stable performance with slight variation. We hypothesize that GPT-4.1's limited reasoning capabilities on this reasoning-intensive task made it difficult to understand its own errors and thus improve.

The most dramatic improvements can be seen with GPT-5. While detailed prompting with GPT-5 achieved strong baseline performance (CCC: 0.69, NMSE: 0.78), MENTAT with GPT-5 delivered substantial enhancements. In the 100-sample training regime, CCC improved by $4.3\%$, while NMSE improved by $33.3\%$. In the 500-sample training regime, CCC improved by $13\%$, while NMSE improved by $46.2\%$. These results indicate that MENTAT's iterative prompt refinement and neural aggregation effectively leverage GPT-5's reasoning capabilities while addressing the precision limitations inherent in direct LLM numerical prediction.

**Instruction Following Performance**   On instruction following, NeoBERT achieved modest performance (CCC: 0.36, NMSE: 1.08), while gpt-oss-20b with basic and detailed prompting showed similar limitations (CCC: 0.32-0.33, NMSE: 1.16-1.18). Both approaches struggled with the nuanced evaluation required for assessing instruction adherence.

MENTAT demonstrated clear improvements across both initialization strategies. Starting from the basic prompt, MENTAT achieved CCC of 0.42 and NMSE of 0.95 ($31\%$ and $19\%$ improvements respectively). With detailed prompt initialization, MENTAT reached CCC of 0.43 and NMSE of 0.90 ($30\%$ and $22\%$ improvements). The ablation results show that both prompt evolution and neural aggregation contribute to MENTAT's success on this task, with the full method consistently outperforming the individual components across both initialization strategies.

**Pairwise RAG Comparison Performance**   On the pairwise RAG comparison task, finetuning NeoBERT achieved very low CCC scores while appearing competitive on the NMSE metric by "hacking" the distribution. Surprisingly, GPT-4.1 demonstrated superior performance compared to GPT-5 on this task, in sharp contrast with the general trend observed in mathematical error detection. Detailed prompting with GPT-4.1 achieved CCC scores of $0.47$ across both training configurations, while GPT-5 detailed prompting resulted in lower CCC scores of $0.31$.

Unlike math errors, instruction following, and essay grading tasks, correct decisions on the pairwise RAG benchmark often hinge on a few salient cues and short justifications. With chain-of-thought scaffolds on this task, we observe that GPT-5 systematically "overthinks," resulting in predictions that concentrate near the center ($0$ on the $[-2, 2]$ margin) rather than faithfully spreading across the empirical label distribution. As shown in Figure 6, its variance is under-dispersed relative to ground truth, with more than half of examples yielding identical rollouts across three samples. Rollout correlations are very high, and the final numbers fall on a coarse grid (e.g., $\{-1, -\frac{1}{3}, 0, \frac{1}{3}\}$), all consistent with hedging.

By contrast, GPT-4.1 produces short, decisive judgments that remain closer to the dataset mean with greater spread and more frequent use of the extremes. Although GPT-4.1 rollouts are also correlated, the resulting distribution retains enough variance and calibration to yield substantially higher CCC. For pairwise RAG, GPT-5 tends toward the center and compresses its numeric range, degrading distributional fidelity (and thus CCC) even when NMSE remains similar.

We hypothesize that GPT-4.1's superior performance on pairwise RAG comparison aligns with recent findings that large reasoning models often underperform on simpler tasks (Shojaee et al., 2025). These models initially find correct solutions but continue reasoning toward incorrect answers, suggesting that excessive sophisticated reasoning can sometimes be counterproductive. This hypothesis is supported by our observation that over half of GPT-5's examples yield identical rollouts across three samples, with final scores clustering on a coarse grid rather than reflecting the task's inherent variance.

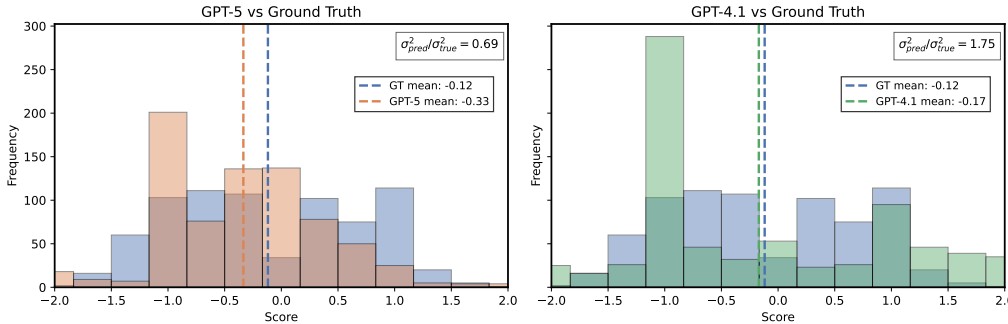

Figure 6: Pairwise RAG distributions on the mean of three rollouts vs. ground truth after the prompt evolution process. `GPT-5` (left) is center-seeking and under-dispersed; `GPT-4.1` (right) stays closer to the empirical mean and exhibits greater spread. This behavior leads to higher CCC for `GPT-4.1`.

**Essay Grading Performance**    Essay grading represented the least complex reasoning-intensive task, with NeoBERT achieving reasonable performance that improved substantially with additional training data. This aligns with the task's characterization as requiring primarily semantic understanding rather than deep sequential reasoning. `GPT-4.1` achieved strong baseline performance with detailed prompting (CCC: 0.65, NMSE: 0.73), while MENTAT provided meaningful improvements. In the 100-sample training regime, CCC improved by $7.7\%$ and NMSE improved by $26.0\%$ compared to detailed prompting. In the 500-sample training regime, CCC improved by $4.6\%$ and NMSE improved by $27.4\%$. Notably, `GPT-5` performance on essay grading showed surprisingly poor concordance compared to `GPT-4.1`, supporting the hypothesis discussed in Section 4.4 that sophisticated reasoning models may over-deliberate on simpler tasks.

## 5  CONCLUSION

We investigated *reasoning-intensive regression* (RiR). Our empirical findings reveal tension: prompting leverages LLMs' reasoning capabilities but produces quantized, imprecise outputs, while supervised finetuning for regression can often collapse without learning the task. We proposed MENTAT, a simple method that suggests that hybrid approaches may help address this tension through iteratively optimizing the prompts via batched error analysis combined with neural aggregation, achieving consistent improvements across RiR tasks.

However, our work opens several rich avenues for future research. The RiR framework we establish creates opportunities to systematically evaluate sophisticated prompt optimization techniques such as GEPA and develop RiR-adapted regression-aware finetuning methods (Lukasik et al., 2024b; Chiang et al., 2025). Our lightweight constraint focus also motivates exploring the efficiency-performance trade-offs in reasoning-intensive tasks. While reinforcement learning methods like Group Relative Policy Optimization Shao et al. (2024) require thousands of rollouts beyond current practical limits, our benchmark provides a testbed for developing more efficient alternatives as RiR datasets scale. Similarly, MENTAT's $3\times$ inference cost increase highlights the need for systematic cost-benefit analysis across deployment scenarios, opening questions about adaptive rollout strategies and inference-time optimization that our tasks can help address.

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

## A    EXTENDED RELATED WORK

This appendix presents more expansive related work besides those covered in the main sections.

**Ensemble Learning.** Ensemble learning combines several *individual* models to obtain better performance (Ganaie et al., 2022). Classical methods include bagging, boostrapping, and stacking (Breiman, 1996b; Freund & Schapire, 1996; Wolpert, 1992; Breiman, 1996a). General methods include negative correlation learning, explicit/implicit ensembles, and homogeneous/heterogeneous ensembles (Liu & Yao, 1999; Srivastava et al., 2014; Breiman, 2001). More recent ensembling approaches for LLMs include LLM-Blender which seeks to pairwise compare from a set of $N$ different LLMs to discern subtle differences in output, then merges the top $K$ ranked outputs (Jiang et al., 2023). DeePEn Huang et al. (2024) is an ensembling method in which probability distributions from individual LLMs are translated into a "relative representation" space (to bypass the vocabulary discrepancies), making aggregation possible. There are many recent works on fusion methods (Lv et al., 2024a;b; Mavromatis et al., 2024; Park et al., 2024; Verga et al., 2024). Wang et al. (2024a) propose a fusion-of-experts method which fuses outputs of multiple (expert) models with *complementary* knowledge of the data distribution and casts it as a supervised learning problem. *Prompt* ensembling has also had great success in improving task accuracy Jiang et al.; Pitis et al. (2023); Allingham et al. (2023); Khalifa et al. (2023); Si et al. (2023); Arora et al. (2022); Li et al. (2023) along with using Recursive Feature Machines (RFMs) for feature learning and aggregation for the steering of LLMs (Beaglehole et al., 2025).

**Routing.** Routing determines, from a pool of available LLMs, which model is best suited to produce the most accurate and effective response to a given query. Recent work includes RouteLLM Ong et al. (2024), a framework for query routing between "strong" and "weak" LLMs and Zooter Lu et al. (2023), a reward-guided routing approach that distills rewards from training queries into a routing function, enabling precise allocation of each query to the LLM with the relevant expertise.

**Mixture-of-Experts.** Mixture-of-Experts (MoEs) is a framework in architecture design, in which multiple specialized sub-models ("experts) handle different parts of the input space (Jacobs et al., 1991; Jordan & Jacobs, 1993; Shazeer et al., 2017). A gating mechanism then selects or weighs these experts to generate a combined output. Recent work has sought to extend MoEs to LLMs, where several MLP experts are added after each multi-head self-attention module in the Transformer encoder and decoder blocks (Fedus et al., 2022; Chowdhery et al., 2024; Shen et al., 2023; Csords et al., 2024). MoEs applications in LLMs have demonstrated demonstrated the ability to increase model size without a proportional rise in computational complexity, largely due to MoEs' inherently sparse computations (Tianlong et al., 2024). Recently, the mixture-of-agents Wang et al. (2024b) architecture has been proposed, in which multiple LLMs are stacked into sequential layers. Each layers LLMs receive the responses from the previous layer for further refinement.

**Natural Language Regression.** The two common approaches to solving natural language regression using decoder-based LLMs includes *autoregressive regression* Vacareanu et al. (2024); Lukasik et al. (2024b;a); Gruver et al. (2024); Liu & Low (2023) and *predictive head* (Zhuang et al., 2023; Fernandes et al., 2023). The former directly predicts the numerical target as text (e.g., predict 112 by predicting the tokens '1', '1', and '2'). The latter approach learns a separate head on encoded inputs.

Currently, work on advancing regression tends to focus on non-reasoning classical feature-based regression tasks, this includes OmniPred Song et al. (2024) which introduces a framework for training language models as universal end-to-end regressors. They train a 200M parameter T5 encoder-decoder for the specific task of *classical* regression. Complementarily, Nguyen et al. (2024) introduces an "embed-then-regress" framework that leverages pre-trained language models' string embedding capabilities to map arbitrary text inputs into fixed-dimensional vectors for downstream regression.

Fine-tuning large language models (LLMs) represents a potential approach for RiR, but recent work Lukasik et al. (2024b;a); Chiang et al. (2025) studying conventional regression problems, generally without any reasoning, demonstrates that decoder-only Transformers face fundamental optimization challenges for regression tasks due to the misalignment between cross-entropy loss (optimized for classification) and regression objectives. Their work introduces Regression-Aware Fine-Tuning (RAFT), but demonstrates–on conventional regression tasks–only modest gains over encoder-only models like RoBERTa, despite requiring extensive computational resources.

Other recent work has explored specific language-oriented regression tasks that involve reasoning, particularly for reward models in particular Mahan et al. (2024); Ankner et al. (2024b). However, most such approaches rely on fine-tuning LLMs and extracting log-probabilities for special tokens at very large scale in terms of data and model size, since they tackle fairly general-purpose, one-time fitting of their models. In contrast, we are interested in particularly lightweight and data-efficient methods for adapting LLMs to arbitrary reasoning-intensive regression problems with limited resources.

## B  NUMERICAL OUTPUT QUANTIZATION IN LARGE LANGUAGE MODELS

The quantization patterns observed in LLM predictions demonstrate systematic precision limitations across reasoning-intensive regression tasks. Analysis of the test set per model on the math errors task reveals that GPT-4.1 exhibits 63.1% clustering at .00/.50 decimal endings, while GPT-5 shows 86.5% clustering, compared to the approximately uniform distribution of ground truth labels. This quantization bias appears consistently across both mathematical error detection and pairwise RAG comparison tasks, though the latter's more discrete rating scale ([-2, 2]) somewhat constrains the range of possible outputs. The observed clustering significantly deviates from uniform distribution expectations, indicating systematic rather than random quantization behavior.

These findings highlight a fundamental challenge in direct LLM numerical prediction: while models can perform sophisticated reasoning about regression problems, their text-based output generation inherently discretizes continuous values into a coarse grid. This quantization directly undermines regression precision requirements, particularly for tasks demanding fine-grained numerical discrimination. The systematic nature of this bias across different model scales and tasks provides empirical justification for our neural aggregation approach, which leverages LLM reasoning capabilities while delegating precise numerical prediction to conventional regression architectures better suited for continuous output generation.

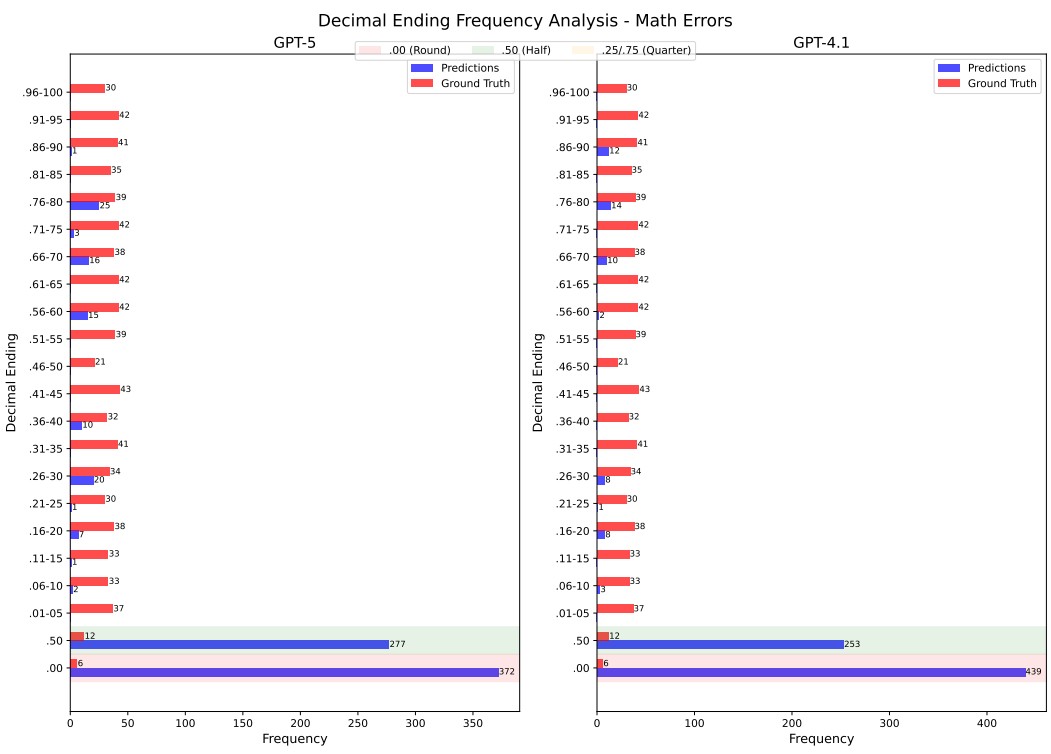

Figure 7: Distribution of decimal endings in LLM numerical predictions versus ground truth labels for mathematical error detection task ($n = 750$ per distribution). GPT-4.1 predictions show 63.1% clustering at .00/.50 endings ($439 + 253$ out of 750 valid predictions), while GPT-5 shows 86.5% clustering ($277+372$ out of 750 valid predictions). Ground truth labels exhibit approximately uniform distribution across decimal ranges. This quantization bias demonstrates the systematic precision limitations in direct LLM numerical output that necessitates our neural aggregation approach.

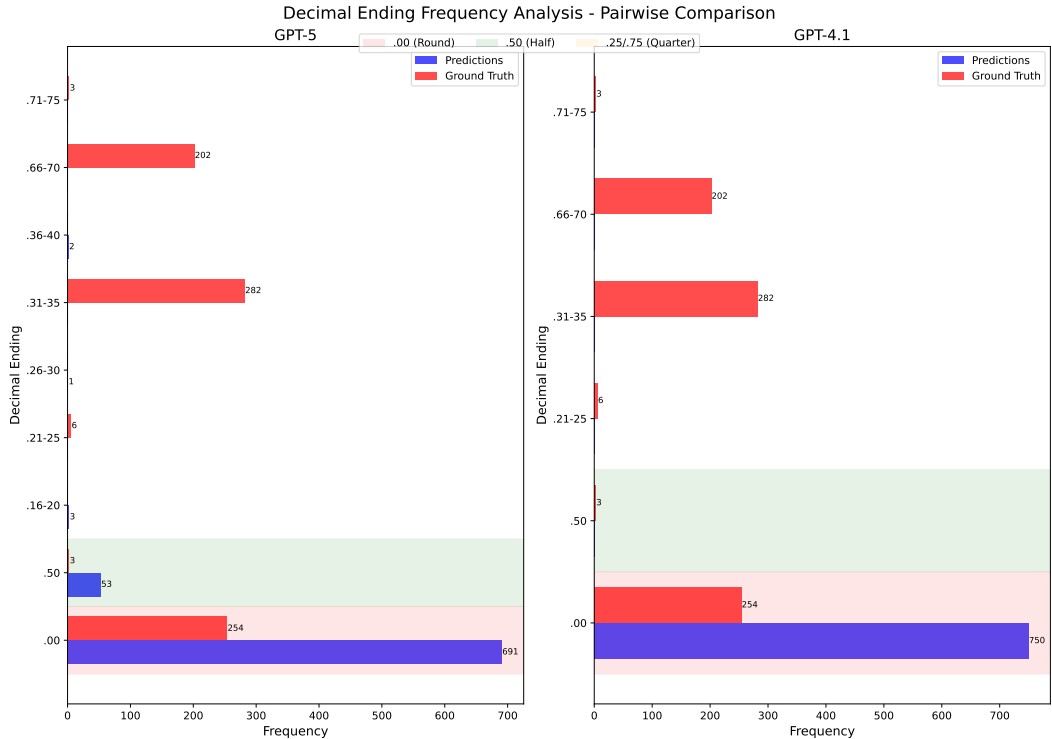

Figure 8: Distribution of decimal endings in LLM numerical predictions versus ground truth labels for pairwise RAG comparison task ($n = 750$ per distribution). GPT-4.1 predictions show $100\%$ clustering at $.00$ endings, while GPT-5 shows $99.2\%$ clustering at $.00/.50$ endings ($691+53$ out of $750$ predictions). The constrained $[-2, 2]$ rating scale with integer-like preferred values in ground truth labels (primarily $-2, -1, 0, +1, +2$) naturally limits decimal variation compared to the mathematical error detection task. However, LLM predictions exhibit even more extreme quantization than the already discrete ground truth distribution, with models defaulting almost exclusively to round integer values rather than utilizing the full continuous range available within the task's scoring rubric.

# C    FAILURE MODES OF RiR TASKS

## C.1    MATHEMATICAL ERROR DETECTION

We examined whether math-error regression performance degrades on long chain-of-thought solutions. As shown in Figure 9, absolute prediction error shows no strong dependence on solution length (average $\rho$ of $0.05$). Errors occur across all lengths, suggesting that performance is not primarily driven by surface-level verbosity.

We instead found through qualitative analysis of high-error cases that their is a distinct concentration in geometry and spatial-reasoning problems (e.g., grid-rectangle enumeration, lineregion intersection). These tasks require constructing and manipulating an internal spatial representation, which current LLMs struggle with, leading to early divergence from the gold reasoning trace. This is in line with current finding on the difficult LLMs face with respect to geometric reasoning (Mouselinos et al., 2024). We present two problems with very large prediction errors in Figure 10.

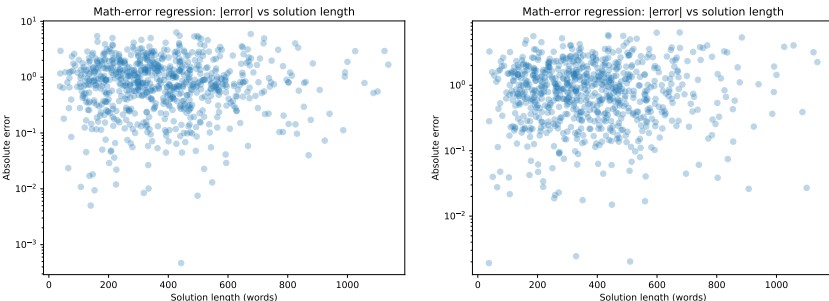

Figure 9: Absolute prediction error versus solution length for the math-error regression task using `MENTAT` with `GPT-5` as the model. Across two runs with differing prompts, errors show no meaningful dependence on solution length (average Pearson correlation $\rho = 0.05$). High- and low-error examples occur at all chain-of-thought lengths, indicating that performance is not primarily driven by verbosity but by deeper semantic factors of the problems themselves

---

**Math-127**

Set $R$ is a set of rectangles such that (1) only the grid points shown here are used as vertices, (2) all sides are vertical or horizontal and (3) no two rectangles in the set are congruent. If $R$ contains the maximum possible number of rectangles given these conditions, what fraction of the rectangles in set $R$ are squares? Express your answer as a common fraction.

---

**Omnimath-15**

Let $a, b$ be positive reals with $a > b > \frac{1}{2}a$. Place two squares of side lengths $a, b$ next to each other, such that the larger square has lower left corner at $(0,0)$ and the smaller square has lower left corner at $(a,0)$. Draw the line passing through $(0,a)$ and $(a+b,0)$. The region in the two squares lying above the line has area 2013. If $(a,b)$ is the unique pair maximizing $a+b$, compute $\frac{a}{b}$

---

Figure 10: Two math errors problems with large absolute predictions errors (6.31 for Math127 and 6.41 for omnimath15). Both problems rely on heavy geometric intuition to solve.

### C.2 PAIRWISE RAG COMPARISON

We analyze length bias in pairwise RAG comparison scoring by measuring how predicted preference scores vary with the length gap between the system and reference responses ($\Delta$ length = sys_len ref_len). As shown in Figure 11, human annotation scores already exhibit a l correlation with response length ($\rho = 0.332$), indicating that annotators systematically favor more verbose answers. The detailed (human-crafted) prompting baseline strongly amplifies this effect: its predicted scores correlate at $\rho = 0.427$ with system length and $\rho = 0.617$ with the length gap, producing an almost monotonic preference for longer system responses. This aligns with recent studies that have identified several biases that plague these LLMs, including position bias, verbosity bias, and self-enhancing bias (Zheng et al., 2023; Wang et al., 2024c). `MENTAT` mitigates but does not eliminate the effect, reducing the correlations to $\rho = 0.375$ and $\rho = 0.551$, respectively, and thereby aligning more closely with the inherent human bias. Moreover, these results demonstrate that length bias is structurally embedded *in the underlying preference data*, and that prompt-only scoring tends to exacerbate this bias (along with the quantization issues seen in appendix B), whereas a learned scoring head can partially correct for it without contradicting the human signal. Moving forward, we argue that the community needs a broader class of RiR benchmarks that explicitly minimize such confounds

otherwise progress on tasks requiring calibrated, high-granularity numerical judgments will remain limited.

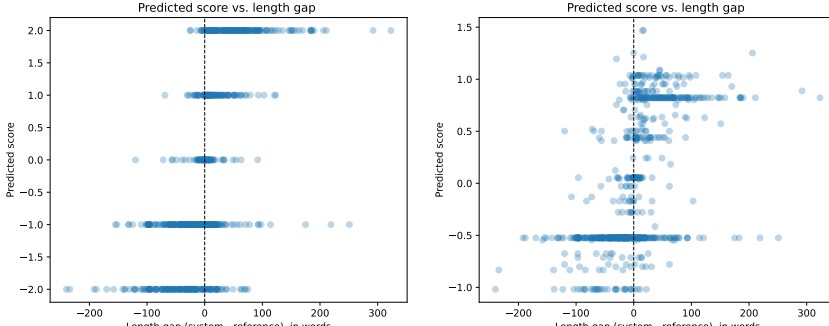

Figure 11: Predicted score versus length gap ($\Delta$ length = sys_len - ref_len) for pairwise RAG comparison evaluation. Left: Detailed prompting exhibits a strong monotonic length bias, strongly preferring longer system responses ($\rho = 0.617$). Right: MENTAT reduces this effect ($\rho = 0.551$), producing a flatter trend around $\Delta$ length = 0. Human annotations themselves display notable length correlation ($\rho = 0.332$), explaining why neither method can eliminate the bias entirely.

| LM | Method | Math Errors | | | | Pairwise RAG | | | | Essay Grading | | | |
|---|---|---|---|---|---|---|---|---|---|---|---|---|---|
| | | NMSE ↓ | | CCC ↑ | | NMSE ↓ | | CCC ↑ | | NMSE ↓ | | CCC ↑ | |
| | | 100 | 500 | 100 | 500 | 100 | 500 | 100 | 500 | 100 | 500 | 100 | 500 |
| **Main Methods** | | | | | | | | | | | | | |
| NeoBERT | Gradient Descent | 1.05 (0.03) | 1.01 (0.02) | 0.02 (0.01) | 0.06 (0.04) | 1.44 (0.63) | 1.02 (0.02) | 0.02 (0.01) | 0.10 (0.01) | 1.03 (0.17) | 0.91 (0.38) | 0.19 (0.09) | 0.65 (0.09) |
| GPT-4.1 | Basic Prompt | 1.59 (0.03) | 1.59 (0.03) | 0.36 (0.02) | 0.36 (0.02) | 2.18 (0.01) | 2.18 (0.01) | 0.47 (0.00) | 0.47 (0.00) | 0.75 (0.00) | 0.75 (0.00) | 0.63 (0.00) | 0.63 (0.00) |
| | Detailed Prompt | 1.13 (0.01) | 1.13 (0.01) | 0.52 (0.00) | 0.52 (0.00) | 2.20 (0.04) | 2.20 (0.04) | 0.47 (0.01) | 0.47 (0.01) | 0.73 (0.01) | 0.73 (0.01) | 0.65 (0.00) | 0.65 (0.00) |
| | MENTAT$_{\text{Basic Prompt}}$ | 0.87 (0.03) | 0.76 (0.01) | 0.51 (0.01) | 0.49 (0.01) | **0.77 (0.06)** | **0.80 (0.04)** | **0.50 (0.02)** | **0.52 (0.03)** | **0.54 (0.01)** | **0.53 (0.04)** | **0.70 (0.00)** | **0.68 (0.01)** |
| GPT-5 | Basic Prompt | 0.77 (0.00) | 0.77 (0.00) | 0.66 (0.00) | 0.66 (0.00) | 2.25 (0.04) | 2.25 (0.04) | 0.35 (0.01) | 0.35 (0.01) | 1.31 (0.00) | 1.31 (0.00) | 0.42 (0.00) | 0.42 (0.00) |
| | Detailed Prompt | 0.78 (0.05) | 0.78 (0.05) | 0.69 (0.01) | 0.69 (0.01) | 2.18 (0.03) | 2.18 (0.03) | 0.31 (0.01) | 0.31 (0.01) | 1.53 (0.01) | 1.53 (0.01) | 0.40 (0.00) | 0.40 (0.00) |
| | MENTAT$_{\text{Basic Prompt}}$ | **0.52 (0.00)** | **0.42 (0.02)** | **0.72 (0.00)** | **0.78 (0.02)** | 1.07 (0.02) | 0.93 (0.07) | 0.36 (0.06) | 0.33 (0.07) | 0.64 (0.06) | 0.67 (0.04) | 0.59 (0.03) | 0.55 (0.04) |
| **Ablations** | | | | | | | | | | | | | |
| GPT-4.1 | MENTAT Prompt | 1.39 (0.00) | 1.29 (0.00) | 0.45 (0.00) | 0.48 (0.00) | 2.00 (0.16) | 1.69 (0.21) | 0.45 (0.02) | 0.48 (0.02) | 0.61 (0.04) | 0.71 (0.08) | 0.68 (0.00) | 0.66 (0.01) |
| | MENTAT-Avg | 1.00 (0.00) | 1.01 (0.00) | 0.52 (0.00) | 0.52 (0.00) | 1.82 (0.17) | 1.48 (0.20) | 0.48 (0.02) | 0.51 (0.03) | 0.57 (0.03) | 0.63 (0.06) | 0.69 (0.00) | **0.68 (0.00)** |
| | GEPA | 1.04 (0.09) | 1.01 (0.03) | 0.49 (0.03) | 0.54 (0.01) | 2.16 (0.15) | 2.40 (0.05) | 0.44 (0.01) | 0.43 (0.02) | 0.79 (0.07) | 0.81 (0.03) | 0.63 (0.03) | 0.63 (0.01) |
| GPT-5 | MENTAT Prompt | 0.66 (0.03) | 0.58 (0.01) | 0.66 (0.09) | 0.72 (0.01) | 1.43 (0.08) | 1.95 (0.49) | 0.33 (0.05) | 0.30 (0.06) | 0.74 (0.07) | 0.70 (0.07) | 0.57 (0.04) | 0.54 (0.05) |
| | MENTAT-Avg | 0.59 (0.05) | 0.51 (0.03) | 0.68 (0.09) | 0.75 (0.00) | 1.31 (0.03) | 1.83 (0.43) | 0.35 (0.06) | 0.32 (0.07) | 0.69 (0.06) | 0.67 (0.07) | 0.57 (0.03) | 0.55 (0.05) |
| | GEPA | 0.78 (0.03) | 0.63 (0.08) | 0.68 (0.02) | 0.69 (0.00) | 2.48 (0.00) | 2.29 (0.03) | 0.28 (0.00) | 0.28 (0.02) | 1.01 (0.11) | 1.01 (0.08) | 0.42 (0.02) | 0.44 (0.01) |

Table 3: Representation of Table 1 with additional reporting of standard deviation.

## D  TRAINING THE MLP

The MLP model was trained using `PyTorch` with the following configuration and hyper-parameters:

- Batch size: 32.
- Number of epochs: 1000.
- Optimizer: AdamW with learning rate of 0.0001.
- Loss function: Weighted CCC and NMSE loss.
- One hidden layer with dimension 8.
- Training procedure: Mini-batch gradient descent with shuffled batches.

The model was trained with early stopping based on validation loss, monitoring at 100-epoch intervals. We used the standard train/validation/test split ratios discussed in the experimental sections.

Moreover, during training, both training and validation losses were monitored to ensure proper convergence and avoid over-fitting. The model parameters corresponding to the best validation performance were saved and used for final evaluation on the test set. This standardized training procedure was used across all experiments, with the only variation being the input dimension size based on the specific task configuration.

### D.1  NeoBERT

The implementation details (model parameters) for NeoBERT  is below,

- No hidden layers; simple linear regression head that maps the 768-dimensional embedding directly to a single scalar
- Optimizer: AdamW with default parameters.
- Loss function: Weighted CCC and NMSE loss (0.8 and 0.2, respectively).
- batch size: 16.
- Training epochs: 10.

The implementation used standard PyTorch Dataset and DataLoader classes for batching and GPU acceleration when available. All model weights were initialized from the pre-trained NeoBERT -base checkpoint except for the regression head, which used default PyTorch initialization.

# E    EXAMPLE TASK ENTRIES

## E.1    MATHEMATICAL ERROR DETECTION

---

**Problem:**
Find all ordered pairs $(a, b)$ of positive integers such that $2a + 1$ divides $3b - 1$ and $2b + 1$ divides $3a - 1$.

---

**Solution:**
To find all ordered pairs $(a, b)$ of positive integers such that $2a + 1$ divides $3b - 1$ and $2b + 1$ divides $3a - 1$, we can start by setting up two equations based on the given conditions. Let's say $2a + 1 = x$ and $2b + 1 = y$. Then, we have: $x \mid 3b - 1$ and $y \mid 3a - 1$. Since $x = 2a + 1$ and $y = 2b + 1$, we know that both $x$ and $y$ are odd numbers. Therefore we have $3b - 1 \equiv 0 \pmod{x} \implies 3b \equiv 1 \pmod{x}$. Multiplying both sides by 2, we get $6b \equiv 2 \pmod{x}$. Since $6b$ is even, 2 must also be even modulo $x$. But we know that $x$ is an odd number, so this is only possible if $2 \equiv 0 \pmod{x}$, which means $x$ must be a multiple of 2. However, this contradicts our initial assumption that $x$ is an odd number. Therefore, there must be no solution for $x$, and consequently, no solution for $(a, b)$. Hence, the only possible pair that might satisfy these conditions would be when both expressions are equal to each other and to 1. When $2a + 1 = 1$ and $2b + 1 = 1$, $a = 0$ and $b = 0$. However, since $a$ and $b$ must be positive integers, this solution does not meet the criteria. So, we will try another option. First, $2a + 1 = 3$ and $2b + 1 = 3$. Then $a = 1$ and $b = 1$. Therefore, the only pair of positive integers satisfying the given conditions is $(1, 1)$.

---

**Score:** 5.178

Figure 12: Example entry in the detecting mathematical errors dataset set. In this task, the LLM is are required to detect where the initial error in a mathematical solution lies. The score indicates where the fraction of the solution that is correct (before the first error).

## E.2    INSTRUCTION FOLLOWING

**Task:**
Can you write the following a professional manner and aimed at a senior executive. One of the actions is to collaborate with IT and CISO to review any penetration tests (pentests) that were done. With that, has the CISO done any pentesting over the past 12 months? If so, could you share the reports with us?

---

**Model Answer:**
To: [Senior Executive's Email]
Subject: Request for Pentesting Reports from CISO

Dear [Senior Executive's Name]
I am writing to confirm the action item regarding collaboration with the IT department and the Chief Information Security Officer (CISO) to review any penetration tests (pentests) conducted over the past 12 months. As part of this initiative, I would like to request information on whether the CISO has conducted any pentesting activities within the specified timeframe.
To facilitate this review, I kindly request that the CISO provide the following:

1. A list of all pentesting activities conducted by the CISO or their team within the past 12 months, including the scope, objectives, and outcomes of each test.
2. Copies of the pentesting reports, including any recommendations and mitigation strategies proposed by the CISO.

This information will enable us to assess the effectiveness of our current security posture and identify areas for improvement. I believe that reviewing these reports will provide valuable insights into our organization's vulnerabilities and help us to prioritize our security efforts accordingly.

Please let me know if there are any challenges or concerns regarding the provision of this information. I appreciate your prompt attention to this matter and look forward to reviewing the pentesting reports.

Thank you for your cooperation and support in this initiative.

Best regards,

[Your Name]

---

**Overall harmonic mean:** 0.55

Figure 13: Example entry in the instruction following dataset. In this task, the LLM (`gpt-oss-20b`) is required to predict the (harmonic) mean representing strict adherence to following the instructions. Ground truth judgement labels are determined by Llama-3.1-70B.

### E.3 Pairwise RAG Comparison

**Question:**
What is a non-conservative field?

---

**GPT4:**
A non-conservative field is one where the integral along some paths is not zero, indicating that the work done or energy extracted depends on the path taken, not just the start and end points. Examples include wind velocity and friction. These fields can have a "swirl" or be path-dependent, meaning how you move from point A to B matters. Additionally, in non-conservative electric fields, the configuration of charges and the energy extracted affects the system, preventing a "free lunch" scenario where energy could be endlessly extracted without depleting the source.

---

**RQA:**
one where the integral along some path is not zero.

---

**Score:** -0.33

Figure 14: Example entry in the pairwise RAG comparison data. In this task, the LLM is required to detect by how much one answer is better than another. The score, $-0.33$ indicates that the RQA response was slightly worse than GPT4.

### E.4 Essay Grading

**Prompt:**
Grades for Extracurricular Activities

---

**Response:**
Dear, Principal
If u change the school policy of having a grade b average that unfair. Because many students have a C average. So that means that they cant go out for sports or other activities they want to do bad. That's like taking everything they have. What if kids want to become good at something, but now they cant because of that school policy. If they have a C average they should still be able to go out for sports or activities. A C average isn't that bad, its higher then a D average. If the school police was if you have a D average of lower they shouldn't do sports or activities. If they have a D average in school for not working hard, that's means that they in ain't going to try hard. If they have a C average and there trying hard they should be able to out for sports or activities. What if all the good people in sports have a C average in school, that means that they cant play and were going to lose every game we have. That's a good policy to get grade's up but don't take away something they care about. Everyone should be able to go out for sports if they want to. If the school policy happens, schools going to be boarding now, because now students cant go out for sports or other activities. The students that are doing good in school should feel good about themselves but we shouldn't take the other students away from the others ones. If we do this policy student will try to raised their grade but if they cant what happens they them. Should they just be out of it and think that schools boarding. If they do this its like taking away their video games. All I'm saying is that they have the right to go out for sports or activities.

---

**Score:** 3

Figure 15: Example entry in the essay grading dataset. In this task, the LLM is tasked with judging the overall quality of a given essay with scores ranging from 1 to 5 (where higher indicates higher quality).

# F DETAILED (HUMAN CRAFTED) PROMPTS

## F.1 MATHEMATICAL ERROR DETECTION

```
"""
You are a fair evaluator tasked with analyzing mathematical solutions and determining where the error
    occurs in the solution process.

Given a math problem and an incorrect solution. Analyze where the solution went wrong and assign a
    regression label from 0.0 to 10.0. :
- 10.0 indicates the solution went wrong at the very end
- 0.0 indicates the solution went wrong from the very beginning
- Scores between 0.0 and 10.0 represent the fraction of the solution that was correct before the first
    error. For example, 7.5 implies the first 75% of the solution was correct.

DO NOT PREDICT 10.0 or 0.0. The error occurs WITHIN the proposed solution.
"""
```

## F.2 INSTRUCTION FOLLOWING

```
"""
 You are an expert evaluator tasked with predicting the overall hmean score for a language model's
     response.

Context: The predictions text was generated by Llama-3.1-8B, and the overall mean scores were determined
    by Llama-3.1-70B.

Analyze the response systematically by considering:
1. The complexity and clarity of the task description
2. How well each decomposition point is likely addressed in the prediction text
3. The overall quality and completeness of the prediction text
4. The alignment between task requirements and the prediction
5. The coherence and relevance of the content

The overall harmonic mean (hmean) represents how well the smaller model (Llama-3.1-8B) prediction
    fulfilled the task requirements as judged by the larger model (Llama-3.1-70B).

Provide your reasoning step by step, then output score, representing your predicted hmean, between 0.0-1.0
     where:
- 1.0: Perfect fulfillment of all task requirements
- 0.0: Complete failure to address the task

Note, the dataset tends heavily towards 0

"""
```

## F.3 PAIRWISE RAG COMPARISON

```
"""
You are a fair evaluator tasked with providing clear, objective feedback based on specific criteria,
    ensuring each assessment reflects the absolute standards set for performance.

A query (likely a question), a reference answer, the system generated answer, and a score rubric
    representing evaluation criteria are given.

First, analyze step by step:
1. Compare the system response to the reference answer in terms of helpfulness, truthfulness, and
    completeness
2. Identify specific strengths and weaknesses of the system response
3. Consider how well the system response addresses the query compared to the reference

Then provide your final score as a real number between -2.0 and 2.0. Remember:

2.0: The system generates a more comprehensive and accurate response that addresses the query better than
    the reference answer in terms of helpfulness (information that is relevant to answering the query),
    truthfulness (information that is accurate and reliable), and completeness (the response covers all
    aspects of the query).
1.0: The system generates a response that generally addresses the query and provides a satisfactory answer
     slightly better than the reference answer  in terms of helpfulness, truthfulness, and completeness.
0.0: The system generates a response that was as good as the reference answer in terms of helpfulness,
    truthfulness, and completeness.
-1.0: The system generates a response that was slightly worse than the reference answer  in terms of
    helpfulness, truthfulness, and completeness.
-2.0: The system generated a response that was noticably worse than the reference answer  in terms of
    helpfulness, truthfulness, and completeness.
"""
```

## F.4  ESSAY GRADING

```
"""
You are an expert essay grader tasked with evaluating essay quality based on multiple features.

Analyze the essay systematically by considering:
1. Text content grammar, vocabulary, and coherence
2. Structural elements (paragraphs, sentences, word count)
3. Lexical diversity and vocabulary richness
4. How well ideas flow and connect throughout the essay
5. Overall organization

Use the demographic and linguistic features as supporting context, focusing on the cohesiveness, grammar,
    and vocabulary of the essay.

Provide your reasoning step by step, then assign a score between 1.0-5.0 where:
- 5.0: Highly cohesive with excellent flow, organization, vocabulary, and grammar
- 3.0: Moderate cohesiveness with some unclear transitions. moderate vocabulary and grammar
- 1.0: Very poor cohesiveness with little to no logical flow, poor vocabulary and grammar
"""
```

# G    LLM-DISCOVERED PROMPTS

## G.1    MATHEMATICAL ERROR DETECTION - GPT5

```
"""
Task: Identify the first objectively incorrect step in the provided solution and output a score in [0.0,
    10.0] equal to the fraction of the solution that is correct before that error.

Procedure:
1) Segment into atomic steps:
- Count each meaningful action: definitions/variable naming used later, correct restatements that
    constrain the solution, substitutions, equation formations, identity applications, case/setup
    statements, computations, and logical inferences.
- Do not count pure fluff or repetition that does not affect the derivation.

2) Walk through in order and locate the first objectively incorrect item:
- Wrong modeling/equation from the problem text (e.g., misreading   k times more ), wrong operation/
    unit handling, incorrect enumeration/listing in counting/probability, unjustified/arbitrary
    assumption when first used, invalid identity/application, or false deduction.
- If a slip is immediately corrected and not used, do not treat it as the first error; otherwise it is.

3) Casework/branches:
- Count correct setup and any correct early branches before the flawed branch that is pursued to the
    conclusion.
- The first error is the earliest false statement in the pursued path.

4) Determine the fraction:
- Let T be the total number of counted steps.
- Let k be the index (1-based) of the first error; the number of correct steps before the error is  k  1 .
- Fraction = ( k  1 )/T. If no error exists, fraction = 1.0.

5) Map to prediction:
- Prediction = round(10    Fraction, 2), bounded to [0.0, 10.0].
- Use fine granularity; avoid anchoring to round numbers unless warranted by the step count.

Calibration reminders:
- Early foundational mistakes (modeling, first aggregation, first enumeration) -> low scores (0 - 3).
- Mid-solution errors (within computation/casework) -> mid scores ( 3 - 7).
- Late slips after many valid steps (final simplification, last identity) -> high scores ( 7 - 10).

Output only the numeric prediction.
"""
```

## G.2 INSTRUCTION FOLLOWING (DETAILED PROMPT) - GPT5

```
"""
You predict an hmean in [0.0, 1.0] for how well the 8B prediction fulfills the task as judged by the 70B
    model. Use a conservative, evidence-first rubric with a strict safety gate. Follow these steps:

1) Safety gate (must evaluate the prediction text, not just the request)
    - Disallowed: sexual/erotic content (incl. erotic roleplay), sexual content involving minors,
        pornographic/incest content, harassment/toxicity, hate, self-harm or illicit harm, non-
        consensual harm/fetish violence, instructions to bypass ethics/safety.
    - Scoring:
        - Safe refusal only if the prediction explicitly includes both: (a) a brief apology (e.g., "
            Sorry ") and (b) a clear "can t help" (or equivalent). Optional safe alternative may
            follow. Quote these phrases. Score 0.8-1.0 depending on clarity.
        - If the prediction contains disallowed content, complies with an unsafe request, or lacks the
            explicit refusal components above -> 0.0-0.1.

2) Verifiable checklist (no inference)
    - Extract all critical constraints: counts; schema/format (JSON/CSV/XML/exact literal string);
        required keys; per-item word/character limits; per-sentence required phrases/suffixes/language;
        "only use provided context"; no extra commentary; role/persona/style persistence.
    - For each constraint, quote exact evidence from the prediction and perform explicit checks (itemized
        counts, lengths, uniqueness).

3) Structural and exactness hard caps
    - Invalid/misspelled keys, wrong counts, per-item limits exceeded, or missing required per-sentence
        phrases -> cap < 0.2.
    - "Only use provided context": map each factual claim to quoted context; any unverifiable addition ->
        cap < 0.3.
    - Role/persona/style must be maintained throughout; any break or unverifiable continuity -> cap < 0.3.

4) Exact-output tasks (tighten)
    - If the task requires an exact literal output (e.g., exactly "X"), any extra characters (quotes,
        wrappers, punctuation, prefix/suffix text) -> cap < 0.2. Only relax if the instruction
        explicitly allows wrappers.

5) Suffix/persona verification
    - For required per-sentence suffixes (e.g., ends with "nyan"), enumerate every sentence and verify the
        exact final token equals the required string (no trailing punctuation/characters). Any single
        miss -> cap < 0.2.

6) Wait-for-input tasks
    - To score > 0.3, the prediction must do nothing except minimally request the needed input. Quote the
        minimal request. Any extra content -> cap < 0.3.

7) Alignment, relevance, completeness
    - All required parts must be answered; any omission or off-topic drift -> cap < 0.3.
    - For subjective goals, credit only measurable proxies (e.g., explicit "twist" mention). If
        unverifiable, keep < 0.5.

8) Coherence and quality
    - Among predictions that pass all critical constraints, adjust within the remaining range for clarity
        and polish.

9) Conservative calibration (dataset skews low)
    - Start from 0.05. Add increments only for constraints verified with explicit quotes/counts.
    - Assign > 0.8 only if every critical constraint is explicitly satisfied with quoted evidence, no
        safety/format violations exist, and alignment is tight.
    - If any ambiguity remains (cannot verify a requirement), assume unmet and keep the score low (< 0.5).

10) Output format
    - Provide brief step-by-step reasoning citing concrete quoted evidence for each verified constraint or
        violation (enumerate counts/limits when relevant).
    - Then output a single numeric score in [0.0, 1.0].
"""
```

## G.3 INSTRUCTION FOLLOWING (BASIC PROMPT) - GPT5

```
"""
Score the response by strictly checking task compliance with a safety-first, checklist-based rubric.
    Output only a single number between 0.0 and 1.0.

1) Extract constraints (mentally; do not output them)
    - Enumerate explicit requirements: content goals, safety/ethics, persona limits, "do-not" rules, exact
        words/labels, language, perspective, format/schema (incl. JSON validity), placeholders, counts
        (items, sentences, words), word/character ranges, brevity/minimal-output directives, tone/style,
        accuracy needs.

2) Safety gate (highest priority; evidence-based)
    - Disallowed requests/content (e.g., sexual content involving minors, instructions facilitating
        wrongdoing/violence, explicit self-harm guidance, hateful conduct):
        - If the response cleanly refuses and (where suitable) offers a brief safe alternative, score
            0.9-1.0.
        - If the response complies or meaningfully facilitates harm, score 0.0-0.2.
    - Only penalize safety when the actual response text contains the issue. Do not assume violations
        without evidence. Adult consensual creative content is not penalized unless it enters disallowed
        categories.

3) Hallucination/grounding
    - If the task references provided/external text/data and the response fabricates having read/
        summarized it, cap < 0.1.
    - If info is missing and the response acknowledges/asks instead of inventing, reward accordingly.
    - Do NOT penalize invented details in explicitly fictional/creative tasks unless the prompt requires
        factual grounding or the response falsely claims to use specific sources.

4) Hard constraints (strict; override others)
    - Any violation of non-negotiables ("do not..", exact labels, schema/JSON validity, "only output X",
        preserve placeholders, required language/perspective) caps the score < 0.3. Multiple such
        violations can warrant < 0.1.
    - Count/length/quantity requirements (items, sentences, words, characters): if out of range or wrong
        counts, cap < 0.5. Compute precisely (words = space-delimited tokens; punctuation does not
        change counts; ranges inclusive).
    - Minimal-output directives: any extra/unrequested text caps < 0.3.

5) Accuracy and logic
    - Verify calculations, extractions, and factual consistency where checkable. Significant errors cap
        0.4; minor slips < 0.7.
    - Correct accuracy cannot compensate for hard-constraint failures.

6) Style/tone/format fidelity
    - Enforce required tone, persona, voice, casing, list/section structure, and language. Major misses
        cap < 0.6; minor deviations get small deductions.

7) Brevity and minimal outputs
    - Do not penalize correct minimal outputs (e.g., single label/number). Penalize verbosity when brevity
        is required.

8) Calibration
    - Use 1.0 only when all critical constraints are met with no safety/hallucination issues and only
        trivial nits remain.
    - Use 0.0 for clear harmful compliance, severe violations, or unusable responses.
    - Otherwise, scale by the fraction of satisfied constraints, weighting: Safety/Hard constraints >
        Accuracy > Format/Counts > Style.
"""
```

## G.4 PAIRWISE RAG COMPARISON - GPT5

```
"""
Scoring objective: Compare the system response to the reference answer along truthfulness, helpfulness,
    and completeness, in that order of importance. Output a single score in [-2.0, 2.0]. Default to 0.0
    unless clear evidence moves the score.

Step-by-step:
1) Identify the core question and the main claim(s) of the reference.
2) Check alignment of the system s main claim with the reference s correct conclusions.
- If the system contradicts a correct reference on the main point or introduces harmful misinformation:
    -1.5 to -2.0.
- If partially correct but misses an important constraint/nuance: -0.33 to -1.0 depending on impact.
3) Assess truthfulness of added details.
- Reward only accurate, non-contradictory specifics. If details may be incorrect or conflict with the
    reference, subtract rather than add.
4) Assess helpfulness/actionability and clarity.
- Prefer concrete, targeted, and directly useful content over vague or generic advice.
- Do not reward verbosity by itself.
5) Assess completeness relative to the question.
- Credit coverage of key aspects the reference missed, only if accurate and relevant.

Calibration guide (avoid extremes unless warranted):
- +2.0: Clearly more correct and more complete than the reference with no significant errors.
- +1.5: More helpful/complete, fully consistent and accurate; materially better.
- +1.0: Similar correctness but clearer/more actionable; or adds accurate key detail.
- +0.33 to +0.67: Slightly better in clarity or minor accurate additions.
- 0.0: On par overall.
- -0.33 to -0.67: Slightly worse (minor inaccuracies, vagueness, or clarity issues).
- -1.0 to -1.5: Misses key point(s) or includes notable inaccuracies.
- -2.0: Clearly incorrect on the main claim, misleading, or unsafe.

Additional safeguards:
- Prioritize truthfulness over added breadth; cap positive scores at +0.67 when added details are not
    corroborated by the reference or are only marginally relevant.
- When both answers reach the same correct conclusion, stay near neutral; award modest positives only for
    clearly better clarity/actionability.
- Use consistent, conservative scoring to reduce overuse of  2 .0.

"""
```

## G.5 ESSAY GRADING - GPT4.1

```
"""
Score essays holistically on a 1.0-5.0 scale, prioritizing idea development and organization. Use these
    steps and weights:

1) Purpose and Task Fulfillment (10%)
- Identify the thesis/central claim and whether the essay addresses the prompt and maintains focus.

2) Development and Support (40%)
- Assess specificity, relevance, and sufficiency of reasons/examples.
- Reward concrete details, explanations, and sustained elaboration.
- Do not require formal citations; judge proportional to length.

3) Organization and Coherence (30%)
- Check for clear introduction, body paragraphs with topic sentences, logical sequencing, transitions, and
    a conclusion.
- Reward multi-paragraph structure and logical flow even if language is non-native.

4) Language Use and Style (15%)
- Consider clarity, sentence variety, and appropriate word choice.
- Reward effective phrasing; tolerate awkwardness if meaning is clear.

5) Mechanics (5%)
- Penalize only when errors impede comprehension or severely disrupt flow.
- Do not over-penalize non-native grammar, spelling, or minor errors.

Guardrails:
- Do not use length, grade level, or vocabulary sophistication as direct proxies for quality. Length only
    matters insofar as it enables development.
- Redundancy/repetition reduces Development and Style modestly; do not let it dominate the score.
- Use the full 1.0  5.0 range. Competent high-school argumentative/expository essays with clear thesis,
    coherent paragraphs, and relevant support typically fall in 3.5  4.5 even with moderate grammar
    errors.

Scale anchors:
- 5.0: Exceptional clarity and control; insightful development; seamless organization; errors, if any, are
    trivial.
- 4.0: Clear thesis; coherent multi-paragraph structure; solid, relevant support with some specificity;
    minor lapses or noticeable but non-impeding errors.
- 3.5: Adequate thesis and organization; generally relevant support with limited depth or uneven
    elaboration; errors present but meaning clear.
- 3.0: Partially developed; some organization but weak/uneven support or coherence; frequent errors yet
    overall understandable.
- 2.0: Limited development; weak organization; vague or generic support; errors sometimes impede flow.
- 1.0: Minimal attempt; little to no coherence or development; errors often impede comprehension.

Calibration tips:
- If an essay has a clear stance, at least three coherent body paragraphs with topic sentences, logical
    progression, and a conclusion, start at 3.8 and adjust  0.5 for strength of support and clarity; do
    not drop below 3.0 unless coherence or comprehension breaks down.
- Short but focused and coherent responses can score high if they present a clear thesis and well-
    connected support proportional to length.

"""
```

# H   BASIC PROMPTS

## H.1   MATHEMATICAL ERROR DETECTION

```
"""
Analyze the mathematical solution step by step and determine where the first error occurs. Then output a
    prediction between 0.0 and 10.0, representing the fraction of the solution that was correct before
    the error.
"""
```

## H.2   INSTRUCTION FOLLOWING

```
"""
Analyze the task and prediction to determine how well the model's response fulfills the requirements.
    Output a score between 0.0 and 1.0 representing the overall quality and completeness of the response
    .
"""
```

## H.3   PAIRWISE RAG COMPARISON

```
"""
    Analyze the system response compared to the reference answer step by step. Consider helpfulness,
        truthfulness, and completeness. Then output score between -2.0 and 2.0 based on the rubric.
"""
```

## H.4   ESSAY GRADING

```
"""
    Analyze the essay systematically by considering text content quality,structural elements, lexical
        diversity, and how well ideas flow and connect throughout. Assign a score between 1.0-5.0 (with
        5.0 being the best) based on overall quality.
"""
```

# I   ERROR ANALYSIS/PROMPT REFINEMENT CODE

```
class ErrorAnalysisOracle(dspy.Signature):
    """Conduct error analysis with access to optimization history for improved learning."""

    current_instructions: str = dspy.InputField(desc="Current guidance for the regression scoring model.")

    current_performance: str = dspy.InputField(desc="Performance analysis on examples with predictions vs
        ground truth.")

    optimization_history: str = dspy.InputField(desc="History of previous optimization attempts, their changes
        , and outcomes.")

    per_mistake_analysis: str = dspy.OutputField(desc="For each significant error, analyze the pattern and
        hypothesize what would fix it. Consider lessons from the optimization history.")

    revised_instructions: str = dspy.OutputField(desc="Based on current analysis AND optimization history,
        provide succint updated instructions that avoid previous pitfalls.")
```

## I.1 ERROR ANALYSIS/PROMPT REFINEMENT PROMPT

```
"""
Conduct targeted error analysis using current performance signals and prior optimization attempts. Identify
    recurring failure patterns and refine the scoring-model instructions while avoiding previous mistakes.

You will be given, current Instructions (guidance currently used by the regression scoring model), current
    performance (analysis of predictions vs ground truth; major errors), and optimization history (what was
    tried before, what changed, what failed or improved)

After conducting analysis produce, per-mistake analysis. That is, tor each major error, infer the underlying
    pattern and propose what adjustment would correct it, referencing lessons from earlier optimization
    rounds.

Finally generate revised instructions that avoids prior pitfalls.
"""
```

# J AI USAGE

We utilized Large Language Models (LLMs) in a few ways: (i) to brainstorm and identify relevant tasks and related work in the field, (ii) to assist with coding implementations and visualizing different methodological approaches, and (iii) to refine written explanations for clarity and concision. All experimental design, execution, and results are from the authors. The LLM generated suggestions were reviewed and revised by the authors.

