# OpenReview forum: "Reasoning-Intensive Regression"
_ICLR.cc/2026/Conference — Submitted to ICLR 2026_

### Official Review · Reviewer_2fyW · 2025-11-01

**Soundness:** 2
**Presentation:** 2
**Contribution:** 2
**Rating:** 2
**Confidence:** 4

**Summary:**

The paper proposes “Reasoning-Intensive Regression (RiR)”,  a subset of text-to-score problems that purportedly require numerical reasoning rather than shallow feature extraction. It classifies four different tasks such as math error detection, instruction following, pairwise RAG comparison, and essay grading into continuous targets to form a benchmark, arguing that standard LLM prompting and small-encoder fine-tuning both struggle. The authors also introduce MENTAT, combining batched prompt evolution with a small neural aggregator over multiple rollouts, and advocate CCC (concordance correlation coefficient) alongside NMSE to avoid variance-collapse pathologies.

**Strengths:**

- The paper identifies a failure mode of distance-only metrics (NMSE) in regression based reasoning tasks and motivates CCC to capture agreement

- Presents a simple, lightweight method (MENTAT) that is shown to improve over frozen prompting and a finetuned encoder on the proposed tasks in the evaluation

- Provides an initial, multi-task benchmark with a clear claim that some real-world scoring problems need deeper analysis than typical sentiment/similarity regression

**Weaknesses:**

- Conceptual motivation for “RiR” and the three-level taxonomy is not well motivatied. RiR is described as “fuzzy,” and the levels are explicitly “informal,” with no clear definition on what tasks could be classified this way. This needs to be strengthened further

- Task taxonomy and task definitions for levels is under-justified. Several tasks (e.g., math) are converted into regression by design (predict fraction until first error), which evaluates a surrogate metric rather than native task success. The authors don’t justify why this surrogate is decision-relevant, nor report native metrics (e.g., accuracy, exact match) alongside the other metrics they report. I would love to hear the author side argument on this.

- MENTAT novelty is unclear relative to prior prompt-optimization. The “batch-reflect” prompt evolution over worst-case subsets looks close in spirit to existing prompt-evolution methods; the draft does not provide a thorough, controlled comparison [1] .


- Aggregator/loss choice lacks justification. The order-invariant MLP over rollout stats and the training objective (emphasizing CCC/NMSE) are presented without ablations on alternative aggregators (non-parametric, rank-based) or loss functions.


- It is unclear what role does NeoBERT play and whether it is fair comparison in the light of the experiments. The paper mixes a finetuned encoder (NeoBERT) with frozen LLMs; this crosses regimes. Matched-budget baselines (e.g., LoRA-tuned small LLMs, stronger encoders) are missing, so the relative claims are hard to interpret.

[1] prompterator: Iterate efficiently towards more effective prompts. Sucik et al, 2023

**Questions:**

Task casting: For math and pairwise-RAG, why is the regression surrogate (e.g., fraction until first wrong step) the right target?

---

> ### Author Response · Authors · 2025-12-03
> **Response to Review**
>
> Thank you for taking the time to read this paper. We are glad to hear that you found our use of CCC noteworthy, as well as the construction of our benchmarks, and formulation of our lightweight method, MENTAT.
>
> > “RiR” and the three-level taxonomy is not well motivatied. RiR is described as “fuzzy,” and the levels are explicitly “informal,” with no clear definition on what tasks could be classified this way. This needs to be strengthened further
>
> We concede that our taxonomy is indeed a heuristic guide, rather than a formal classification system. Nonetheless, we are able to support and observe its effects empirically on model behavior.  A clear example of this is NeoBERT's behavior: it achieves CCC 0.65 on Essay Grading (Level 2-adjacent) but collapses to CCC 0.01-0.06 on Math Errors and Pairwise RAG (Level 3), demonstrating that reasoning-intensive tasks resist simpler feature-matching approaches. This empirical gap motivates RiR as a distinct challenge. We provide examples of each tasks in Appendix E.
>
>
> > Task taxonomy and task definitions for levels is under-justified. Several tasks (e.g., math) are converted into regression by design (predict fraction until first error), which evaluates a surrogate metric rather than native task success. The authors don’t justify why this surrogate is decision-relevant, nor report native metrics (e.g., accuracy, exact match) alongside the other metrics they report. I would love to hear the author side argument on this.
>
> We would argue that our metric mirrors what might be used in practice for rubric-based evaluation and error localization,  evaluates deeper analysis in a more fine-grained way than binary classification, and  enables studying calibration and distributional properties. This is why CCC is a far more useful metric. A metric such accuracy or exact match on RiR tasks, where predictions are *continuous* would not be as useful. We argue for this in the text, see section 2 around lines 153-170.
>
> > MENTAT novelty is unclear relative to prior prompt-optimization. The “batch-reflect” prompt evolution over worst-case subsets looks close in spirit to existing prompt-evolution methods; the draft does not provide a thorough, controlled comparison [1] .
>
> MENTAT, while simple, is well positioned for RiR tasks and distinct from other methods . Unlike self-consistency MENTAT works by training an MLP on the statistics of rollouts, and unlike other prompt optimizers, MENTAT is an extremely parallel method that leverages joint reflection on very large batches of errors (i.e., batch error analysis). The proper comparisons against MENTAT are two primary methods: finetuning a recent encoder-only model (e.g., NeoBERT), and prompting a frozen LLM (e.g., GPT4.1, GPT5), both of which this paper does. As an additional baseline, we also test GEPA, a recent prompt optimizer. MENTAT performs better than GEPA, in both NMSE and CCC, on these RiR tasks (see Table 1), while having other attractive properties for regression problems.
>
> > Aggregator/loss choice lacks justification. The order-invariant MLP over rollout stats and the training objective (emphasizing CCC/NMSE) are presented without ablations on alternative aggregators (non-parametric, rank-based) or loss functions.
>
> There are many loss functions or alternative aggregators methods that one *could* choose. We believe our use of NMSE/CC is well motivated (again see lines 153-170) and our choice to aggregate using an MLP is a natural one. We leave the study of other aggregation methods to future work.
>
> >It is unclear what role does NeoBERT play and whether it is fair comparison in the light of the experiments. The paper mixes a finetuned encoder (NeoBERT) with frozen LLMs; this crosses regimes. Matched-budget baselines (e.g., LoRA-tuned small LLMs, stronger encoders) are missing, so the relative claims are hard to interpret.
>
> NeoBERT's role, along with basic/detailed prompting,  is demonstrating RiR's challenge: even with supervised finetuning and favorable data splits (350 train vs 250 for prompt methods), it collapses on reasoning-intensive tasks. It’s also relatively new (published in February of this year).  The goal here is to show  that standard supervised learning fails where LLM reasoning succeeds, motivating our hybrid approach. Adding LoRA-tuned LLMs would be interesting but  does little to push forwards the claims of the paper.
>
> For clarification, when you state “ The paper mixes a fine-tuned encoder (NeoBERT) with frozen LLMs; this crosses regimes, ” what do you mean? If you mean these are two completely different methods for solving RiR tasks, why is this an issue?

---

### Official Review · Reviewer_WmpZ · 2025-11-02

**Soundness:** 4
**Presentation:** 1
**Contribution:** 2
**Rating:** 2
**Confidence:** 2

**Summary:**

This paper studies the problem of reasoning-intensive regression, which requires the LLMs to deduce subtle numerical scores from text. The authors proposes a prompt optimization paradigm by LLM self reflection and showcase the improvement compared to baseline head fine-tuning and prompt hand-crafted fine-tuning methods.

**Strengths:**

The study of reasoning-intensive regression of LLMs is indeed of crucial importance and there lacks work that seriously investigate the limitation of existing LLMs in use cases like LLM as a judge. The direction that goes beyond hand crafted prompt tuning is interesting and appealing.

**Weaknesses:**

1. The benchmarking tasks in Section 2 is rather confusing. Why we choose these tasks as benchmarks for reasoning-intensive regression? There lacks sufficient discussion on the coverage of these tasks and what is the performance of existing either open source or closed source LLMs.

2. The proposed method MENTAT requires additional training and inference burden, there lacks a clear description of the motivation behind this method. In addition, there lacks explanation of what types of data the additional MLP is trained on. What is the input and output of the MLPs? What is the scale of this additional training part upon the original LLMs? What is overall the additional inference cost of current method compared to the standard fixed template of existing LLMs or few shot templates?

3. Disclaimer: I am not familiar with this field of reasoning-intensive regression. But I do think there is huge space to improve on the writing side to make the paper readable for broader audience. I have read many papers, including those in the area I am not familiar with, and this is really one of the few times I found a paper's wordings and technical descriptions make me feel super puzzled almost throughout the paper, making it extremely hard to follow. I would highly recommend adding more relevant equations to explain the tasks and metrics more detailedly, instead of using long sentences and only word descriptions of the methodology. Overall, I do not think this work does a good job in explaining clearly the method proposed.

**Questions:**

See questions in weakness.

---

> ### Author Response · Authors · 2025-11-25
>
> We thank the reviewer for their feedback. We are glad that, despite some confusion, the reviewer finds the study of the topic we introduce "of crucial importance" and our direction "interesting and appealing". We clarify individual items below.
>
> > The benchmarking tasks in Section 2 is rather confusing. Why we choose these tasks as benchmarks for reasoning-intensive regression? There lacks sufficient discussion on the coverage of these tasks and what is the performance of existing either open source or closed source LLMs. [...] Disclaimer: I am not familiar with this field of reasoning-intensive regression.
>
> We would like to clarify that one of our central contributions is defining the problem of reasoning-intensive regression (RiR) and introducing an initial benchmark for it. As such, there is no established field on RiR per se prior to our work.
>
> In Section 2, we motivate and describe four tasks for benchmarking RiR. We foucs on explaining why they combine regression and reasoning, which is the hallmark of RiR problems. For instance, for math errors, we explain that "a model must systematically reason formally about math steps rather than relying on probabilistic heuristics, but it must also be good at estimating relative lengths and inferring the boundaries of the steps in a calibrated way."
>
> The performance of relevant models can be seen in Figure 1, Table 1, and Table 2. The appendices provide further analysis of how models perform in interesting ways on RiR problems: score quantization (Appendix B) and a number of failure modes (in a newly added Appendix C).
>
>
> > There lacks explanation of what types of data the additional MLP is trained on. What is the input and output of the MLPs? What is the scale of this additional training part upon the original LLMs? What is overall the additional inference cost of current method compared to the standard fixed template of existing LLMs or few shot templates?
>
> The goal of the MLP in MENTAT is to aggregate multiple rollouts from the LLM into a single final prediction. To quote the relevant portion of Section 3: "We train a small Multi-Layer Perceptron (MLP) to combine rollout predictions. The aggregator ensures order invariance by sorting rollout predictions, incorporates statistical features (mean, standard deviation, min, max), and is optimized for a combination of the CCC and NMSE loss functions."
>
> So, to clarify, the input is a very small vector that summarizes the three rollouts from the LLM (e.g., their mean, std, min, max, and sorted values). The output is a single scalar prediction.
>
> For Phase 2, we generate 3 rollouts per sample, so MLP training uses 50×3=150 or 250×3=750 input vectors. The cost of training or using the MLP is negligible. Generating three rollouts is the primary source of inference cost. We have expanded our discussion of computation cost in section 4.3.1 (lines ~371-427).
>
>
> > I would highly recommend adding more relevant equations to explain the tasks and metrics more detailedly, instead of using long sentences and only word descriptions of the methodology. Overall, I do not think this work does a good job in explaining clearly the method proposed.
>
> We are eager to improve clarity for any components of the paper that should be stated formally. Currently, Section 2 uses mathematical notation as appropriate to define our metrics (NMSE, CCC) and task formulations, but we would be happy to offer a more formal treatment for any portion of the work that the reviewer specifies.

---

### Official Review · Reviewer_i7ig · 2025-11-03

**Soundness:** 3
**Presentation:** 2
**Contribution:** 2
**Rating:** 4
**Confidence:** 4

**Summary:**

The paper introduces Reasoning-Intensive Regression (RiR), an interesting category of tasks where models performs detailed reasoning and output precise numeric values. It presents a benchmark of tasks such as math error detection, essay grading, showing that standard fine-tuning often collapses to mean predictions, while prompting large models yields reasoning but poor numeric calibration. To address this, the authors propose MENTAT, a lightweight method that first optimizes prompts, then does multiple rollouts, get regression outputs from them, and combine them using an external MLP. The paper is well written however, at times it felt overly verbose. The methodology itself is sound, but noveltywise, it felt like a mixture of prompt optimization and self consistency. More importantly, based on the framing of the task, the paper misses important baselines to compare against to showcase the efficacy of MENTAT.

**Strengths:**

1. This paper introduces Reasoning-Intensive Regression, which is a timely task to investigate into when LLMs are sufficiently powerful to finish important tasks.
2. The four different benchmark datasets are quite interesting to be framed for reasoning-intensive regression, although I believe the task choices could be even more well thought out.

**Weaknesses:**

1. MENTAT seems like using self-consistency on top of prompt optimization. I am not sure whether about the contribution in that aspect, as it is well known that both of them should substantially boost performance.
2. "optimizing prompts for RiR tasks has the fairly unique property that the patterns across examples are at least as important as the per-example error." - I believe this is a general theme of most automated prompt optimization methods. More importantly, authors claim that "MENTAT’s prompt evolver is centered around asking the LLM to jointly reason about tens of mistakes at once". I am not sure it is accurate given methods like APO/ PE2 / TextGrad has already covered those areas.

3. I think the authors need to cover more baeslines here. Modern Prompt optimization methods should definitely be covered as baselines to prove the efficacy of the suggested approach. Also, the original self consistency can be a good baseline to compare against.

[1] Pryzant R, Iter D, Li J, Lee YT, Zhu C, Zeng M. Automatic prompt optimization with" gradient descent" and beam search. arXiv preprint arXiv:2305.03495. 2023 May 4.
[2] Ye Q, Axmed M, Pryzant R, Khani F. Prompt engineering a prompt engineer. arXiv preprint arXiv:2311.05661. 2023 Nov 9.

**Questions:**

Please check the weaknesses. The claims need to be substantiated and coverage of prompt optimization methods also needs to be improved, although I understand that is not the primary focus of the problem itself.

---

> ### Author Response · Authors · 2025-11-25
> **Referencing APO/PE2/TextGrad + Baseline Update Addition**
>
> We thank the reviewer for taking the time to offer feedback on our work, especially related to robustifying the baselines. We were glad to hear that they found our problem definition of RiR timely, and the set of benchmarks interesting.
>
> > MENTAT seems like using self-consistency on top of prompt optimization. I am not sure whether about the contribution in that aspect, as it is well known that both of them should substantially boost performance.
>
> Though MENTAT is indeed positioned as an exceptionally simple method that greatly advances RiR performance, we would like to disagree with the reduction in this comment. Self-consistency is an approach that works by counting (i.e., finding the most common answer) and typical prompt optimizers differ quite substantially from MENTAT's. Nonetheless, our work is written with the explicit intent of creating and leaving much headroom for future work on this hard new task.
>
> Unlike self-consistency, MENTAT works by training an MLP on the statistics of rollouts. And unlike other prompt optimizers, MENTAT is an extremely parallel method that leverages joint reflection on very large batches of errors (i.e., batch error analysis). We believe both to be novel, but their exceptional simplicity is the point: we are confident that our reviewer would agree that simple methods with surprising effectiveness are the ones which generalize best and tend to have the most impact, in ML, after all.
>
> ## Reasoning Jointly
>
> > “optimizing prompts for RiR tasks has the fairly unique property that the patterns across examples are at least as important as the per-example error." - I believe this is a general theme of most automated prompt optimization methods. More importantly, authors claim that "MENTAT’s prompt evolver is centered around asking the LLM to jointly reason about tens of mistakes at once". I am not sure it is accurate given methods like APO/ PE2 / TextGrad has already covered those areas.
>
> We disagree with the reviewer's understanding. While our goal from this work is not to advance general-purpose prompt optimization, we must note that existing prompt optimizers typically work by seeking to fix individual or very few mistakes and then aggregating later. For example, TextGrad calculates a "gradient" for each mistake (before "averaging") and methods based on self-bootstrapping generally build few-shot examples of individual successes.
>
> Instead, we're interested in showing that as reasoning models improve, a far simpler (and far more parallel) approach of setting models up to conduct _batch_ error analysis across up to tens of examples _simultaneously_ is a highly effective and cost-efficient strategy. This is our intent from calling our method "batch-reflective".
>
> 1) Our focus is on distribution-level patterns (matching score distributions, calibration) rather than just per-example accuracy. We select the worst $\sqrt{n}$ examples specifically to identify systematic biases that affect the entire output distribution.
>
> 2) Our specific design combines: (a) batch reflection with optimization history to avoid cycles, (b) emphasis on calibration patterns across examples, and (c) neural aggregation of the optimized prompts.
>
> **See the updated Section 3 around lines 245-254. We additionally (see below) take the reviewer's comments as indication we need to increase the number of experiments with the state-of-the-art in general prompt optimization. We describe this below.** Thank you so much!
>
> ## More Baselines
>
> > I think the authors need to cover more baselines here. Modern Prompt optimization methods should definitely be covered as baselines to prove the efficacy of the suggested approach. Also, the original self consistency can be a good baseline to compare against.
>
> Thank you for this suggestion,  **we’ve added a prompt optimizer, GEPA, to table 1 (you can also see the full table with standard deviations on pg 22 in the appendix).  We will update Table 2 shortly!** Overall, GEPA on its own is far less effective than MENTAT's complete design, but even on comparing the MENTAT prompt optimization method, it can be seen that in addition to advantages in quality, MENTAT-Prompt enjoys much better parallelism or efficiency tradeoffs (see line 375 in red for expanded discussions of efficiency).
>
> As for original self consistency, it is unclear how that might work considering that the outputs of regression are numerical. Across three rollouts, we may not necessarily have much information for vanilla self consistency. However, we would like to point out that in our results tables the method "MENTAT-Avg" replaces the MLP in our method with an averaging step, which can be considered one simple generalization of self-consistency within the _rest_ of the MENTAT approach.
>
> **Do the above actions address your concerns with the paper? If not, what further clarification or modifications could we make to improve it?**

---

### Official Review · Reviewer_yiZN · 2025-11-03

**Soundness:** 3
**Presentation:** 2
**Contribution:** 2
**Rating:** 4
**Confidence:** 4

**Summary:**

This paper introduces MENTAT, a streamlined framework to improve the LLM's performance on reasoning-intensive regression (RiR) tasks. This is a two stage process, starting with auto prompt optimization, followed by ensemble learning. It only involes inference on frozen LLMs, and does not require large training datasets, making it lightweight and applicable in practical real world settings.

**Strengths:**

- The MENTAT framework automates the prompt engineering process, which can be very useful, especially in complex scenarios like RiR, where traditionally requires extensive amount of human involvement.
- Ensemble learning is an interesting way to take advantage of the probabilistic nature of LLMs, and increase the regression quality.

**Weaknesses:**

- When introducing MENTAT in Sec.3, it lacks important details, specifically authors should explain
  - Phase 1
    - How to select the bottom $\sqrt{n}$ rollouts in the training set. Are the rollouts ranked by a combination of CCC/NMSE? If so, is the ratio constant, and is the ratio the same as the one used for MLP training?
    - How are the prompts structured in `error analysis` and `prompt refinement`? Does the setup vary between models and/or tasks? Should include all the prompts used in the Appendix.
    - Consider include the conversation of a full iteration. This can be added to the Appendix.
- In Sec.4, results lack ablation study and analysis of the MENTAT framework
  - The performance gap between different veresions of MENTAT makes me wonder if the success of MENTAT is primarily driven by the aggregation of the LLM's probabilistic rollout or is it due to a better prompt generated from the automated process. Authors should include ablation study wrt basic/detailed human crafted prompts (HCP), specifically `HCP Avg` and `HCP MLP` in the results, and compare with their MENTAT counter parts to better analyse the framework. For HCP MLP, training should use the same amount of data and HPT as in MENTAT Phase 2.
  - Authors should add a figure for the distribution of `{Var(all rollouts for question i)}` for each task with HCP and MENTAT Prompt. This should help analyse if aggregation is the key in obtaining a better regression performance.
  - To help understand the effect of prompt optimization, authours can include a `NMSE/CCC vs Iteration` plot in the training process, consider increasing the total iterations for long term analysis. As stated in Appendix.C, MLP is very lightweight, and it should be feasible to include plots of both with and without training the aggregator at each iteration.
  - Add variance to Table.1,2 if space allows. If not, this can be included in the Appendix.
- In Appendix.F, include all the LLM discovered prompts for entries that appear in Table.1,2

**Questions:**

- In Sec.2, regarding the dataset and their evalatuion metrics,
  - Detecting Mathematical Errors, why filter out problems with correct solutions that existed in the original dataset? I think this should be included to truly reflect the real world setting.
  - Instruction Following, original paper used binary label for each requirement $r_i$, and labels are averaged to produce the final score. Why change to $s_i \in [0,1]$ for each $r_i$, and adopt harmonic mean?
  - Pairwise RAG Comparison, why not use the evaluation metric from the original paper, namely RAG-QA Arena?
- How much data were used to train MLP?
- Table.1 mentioned that NeoBERT performs better with "1000 training + 500 validation", why not add an additional column to the results?
- Why not use a universal evaluation model (potentially more powerful than the test model) in error analysis and prompt refinement?

---

> ### Author Response · Authors · 2025-11-25
> **Variance Analysis Addition**
>
> Thank you for taking the time to read this paper, we are grateful for your help, especially related to questions of analysis MENTAT further (variance analysis). We were glad to hear that you found the MENTAT framework thoughtful especially in the RiR settings as well as our use of ensemble learning.
>
>
> ## Phase 1 Explanations
>
> > When introducing MENTAT in Sec.3, it lacks important details, specifically authors should explain Phase 1:  **(1)** How to select the bottom $sqrt{n}$ rollouts in the training set. Are the rollouts ranked by a combination of CCC/NMSE? If so, is the ratio constant, and is the ratio the same as the one used for MLP training? **(2)** How are the prompts structured in error analysis and prompt refinement? Does the setup vary between models and/or tasks? Should include all the prompts used in the Appendix.
>
>
> **(1) We have clarified that the $\sqrt{n}$ worst examples are selected based on absolute prediction error in the caption of Figure 4**, which directly identifies samples where the prompt fails most.
>
> **(2)** Appendices E-G contain detailed, basic, and LLM-discovered prompts. **We have added the meta-prompts used for error analysis/prompt refinement to complete the documentation in Appendix I**.
>
> ## Ablations
>
> > In Sec.4, results lack ablation study and analysis of the MENTAT framework. **(1)** The performance gap between different versions of MENTAT makes me wonder if the success of MENTAT is primarily driven by the aggregation of the LLM's probabilistic rollout or is it due to a better prompt generated from the automated process. Authors should include ablation study wrt basic/detailed human crafted prompts (HCP), specifically HCP Avg and HCP MLP in the results, and compare with their MENTAT counter parts to better analyse the framework. For HCP MLP, training should use the same amount of data and HPT as in MENTAT Phase 2. **(2)** Authors should add a figure for the distribution of {Var(all rollouts for question i)} for each task with HCP and MENTAT Prompt. This should help analyse if aggregation is the key in obtaining a better regression performance. **(3)** To help understand the effect of prompt optimization, authors can include a NMSE/CCC vs Iteration plot in the training process, consider increasing the total iterations for long term analysis. As stated in Appendix.C, MLP is very lightweight, and it should be feasible to include plots of both with and without training the aggregator at each iteration. **(4)** Add variance to Table.1,2 if space allows. If not, this can be included in the Appendix.
>
> **(1)**  Table 1 enables decomposition of MENTAT's gains:
> - Detailed Prompt (HCP) → MENTAT Prompt: Shows prompt optimization effect (Phase 1 only)
> - MENTAT Prompt → MENTAT-Avg: Shows multi-rollout averaging effect. We note that if no optimized prompt is found, the default *would be* HCP
> - MENTAT-Avg → MENTAT: Shows MLP aggregation effect over averaging
>
> Consequently HCP Avg (detailed prompt + multi-rollout averaging) would likely perform on par or worse than MENTAT-Avg!
>
> **(2) This is a great suggestion, thank you! We have added add rollout variance distributions comparing HCP vs MENTAT Prompt to show whether optimized prompts produce more or less consistent predictions. See our new paragraph and figure under section 3 providing some analysis of this (line 286-310).** Essentially, to understand the interplay between prompt evolution and neural aggregation, we analyze the distribution of per-question rollout variances for both the detailed (human-crafted) prompt and the MENTAT-evolved prompt across three task. We found that MENTAT prompt evolution consistently reduces rollout variance compared to the detailed prompt baseline for the more reasoning-intensive tasks. For example, achieving a 30% reduction in mean variance on Mathematical Error Detection. This indicates that the evolved prompts do not merely provide noisy signals for the aggregator to smooth; rather, the prompt
>
> **(3)**  We appreciate this suggestion but believe it's less critical given that we use only 3 iterations by design (the paper already explains this is intentional to minimize compute), the comparison between Basic Prompt to MENTAT Prompt already demonstrates Phase 1's cumulative effect, Training MLP at each iteration would require separate train/val splits per iteration, fundamentally changing the method. Currently the MLP is only used after Phase I, and finally the current  additions (GEPA comparison, compute costs, variance analysis), expands the scope, **We will write a note of this being an interesting case study in the appendix.**
>
>
> **(4) The addition of standard deviation to Table 1 can be  found in the appendix (page 22). For Table 2, this was added within the main text.** Thank you!

---

> > ### Author Response · Authors · 2025-11-25
> > **Clarification of Experimental Decisions (Scoring, Mean etc.)**
> >
> > > In Appendix.F, include all the LLM discovered prompts for entries that appear in Table.1,2
> >
> > Appendix G (was previously F) includes representative LLM-discovered prompts for each task demonstrating the qualitative evolution from basic to optimized prompts. Including all prompt variants (multiple runs × multiple tasks × multiple models) would substantially increase appendix length without additional scientific insight. We believe the included examples sufficiently illustrate the prompt optimization patterns. **We are happy to release all prompts in a supplementary code repository with a link in the final paper!**
> >
> > ## QUESTIONS
> > >In Sec.2, regarding the dataset and their evaluation metrics: **(1)** Detecting Mathematical Errors, why filter out problems with correct solutions that existed in the original dataset? I think this should be included to truly reflect the real world setting.
> > **(2)** Instruction Following, original paper used binary label for each requirement $r_i$, and labels are averaged to produce the final score. Why change to $s_i \in [0,1]$ for each $r_i$ , and adopt harmonic mean? **(3)** Pairwise RAG Comparison, why not use the evaluation metric from the original paper, namely RAG-QA Arena?
> >
> > **(1)**  Including correct solutions would require assigning them a score, but neither 0.0 (error at beginning) nor 10.0 (error at end) is semantically appropriate for "no error exists." This would effectively transform the task into classification (correct vs. incorrect) with an arbitrary regression label for the "correct" class, undermining the continuous regression formulation. Our focus is specifically on localizing errors within incorrect solutions, which is the reasoning-intensive component.
> >
> > **(2)** We note the following
> >
> > - Continuous scores [0,1]: The original binary labels (with average) would still make this closer to classification. Probability-like scores from the judge model create a continuous regression target better suited for studying RiR.
> > - Harmonic mean: Arithmetic mean treats all requirements equally, but we believe that instruction following should have an "all-or-nothing" property. Failing one critical requirement should result in a  substantially lower score even if other, much more basic requirements, are satisfied. The harmonic mean penalizes low individual scores more heavily, capturing that violating any constraint is problematic. This is analogous to how F1-score uses harmonic mean of precision/recall.
> >
> > **(3)** RAG-QA Arena produces win/loss/tie outcomes and ELO-style rankings, which are ordinal rather than continuous. Our regression formulation requires continuous scores, so we use the raw human judgment scores (averaged across 3 annotators on a [-2, 2] scale) rather than the Arena's ranking-based evaluation. This better preserves the continuous regression structure essential for studying RiR while using the same underlying human judgments.
> >
> > > How much data were used to train MLP?
> >
> > For prompt optimization methods including MENTAT, we use balanced train/validation splits: 50+50 (100 total) or 250+250 (500 total). Phase 2 generates 3 rollouts per sample, so MLP training uses 50×3=150 or 250×3=750 input vectors. The MLP architecture (8 hidden units) is deliberately lightweight to prevent overfitting at these scales.
> >
> > > Table.1 mentioned that NeoBERT performs better with "1000 training + 500 validation", why not add an additional column to the results?
> >
> > One of the paper's core thesis is that RiR tasks typically have limited training data. This is a defining constraint of ad-hoc applications. Adding a 1500-sample regime would contradict our motivating assumptions thus shifting focus away from the data-efficient methods we propose. Moreover, adding 1500 (across all tasks) is infeasible for some datasets (math errors only has ~1700 total data points). We include the note about NeoBERT's 1500-sample performance on Pairwise RAG (Table 1 caption) specifically to show that encoders can eventually learn with sufficient data (at least on some RiR tasks), reinforcing that one big challenge is the low-data regime.

---

> > > ### Author Response · Authors · 2025-11-25
> > > **Universal evaluation model**
> > >
> > > > Why not use a universal evaluation model (potentially more powerful than the test model) in error analysis and prompt refinement?
> > >
> > > **Are you suggesting using a stronger model (e.g., GPT5) for error analysis/prompt refinement while, potentially, using a weaker model (e.g., GPT-4.1) for predictions?** If so, this is an interesting direction, and one we've tried (!), but introduces additional complexity and cost. Our current design uses the same model for analysis and prediction, ensuring the error patterns identified are relevant to the model that will execute the prompt. When we performed experiments with GPT5 as the error analysis/prompt refining model, but with GPT4.1 performing the predictions on math errors and pairwise RAG, GPT5 was able to find better prompts than GPT4.1 as a prompt refiner, and it did lead to some (small) gains with GPT4.1 as the test model (compared to using GPT4.1 for both). But, for simplicity sake, we stuck to the same model as both the evaluator and predictor.
> > >
> > >
> > > **Thank you again for taking the time to review the paper and providing helpful feedback! Do the above actions address your concerns with the paper? If not, what further clarification or modifications could we make to improve it?**

---

> > > > ### Comment · Reviewer_yiZN · 2025-11-26
> > > > **Re: Universal evaluation model**
> > > >
> > > > I think this deserves a mention in the Conclusion!

---

> > > ### Comment · Reviewer_yiZN · 2025-11-26
> > > **Experiment Scoring**
> > >
> > > (1) Correct answer is a very important aspect when it comes to "detecting mathematical errors". In ProcessBench, they proposed to use -1 label for the correct samples. I agree that this is not "continuous" compare to the incorrect samples with labels in [0, 10], and I doubt it is going to change the success of MENTAT, but I still strongly insist that they should be included for completeness.

---

> > ### Comment · Reviewer_yiZN · 2025-11-26
> > **Prompt Optimization Explanation & Ablation Study**
> >
> > # Prompt Optimization Explaination
> > In Appendix I, please include at least one full example of the **refinement prompt**, not the code snippet. It should look like Appendix F, and please use highlight to clearly indicate the **guiding part** and the **input part**. Right now Appendix I is giving a better idea how this auto prompt tuning works behind the scnene, but the delivery is not optimal.
> >
> > # Ablation Study
> > (1) I understand the current decomposition of MENTAT gains, and I also see the "trend" for HCP Avg < MENTAT Avg and HCP  MLP < MENTAT as HCP < MENTAT Prompt. However, this "trend" is still a hypothesis, and **please back it up with data**. You can include it in the Appendix if the table gets too long. This should give some analysis on the impact of the ensemble aggregator.
> >
> > (2) Please also include rollout variance distribution for **instruction following**
> >
> > (3) A new MLP is trained for every iteration, and you can always use the same train/test split. There are no leakage, since "intermediate" MLP is used only for analysis, and does not affect the actual prompt refinement cycle

---

> > > ### Author Response · Authors · 2025-11-27
> > > **Response to Prompt Optimization Explanation & Ablation Study**
> > >
> > > ## Prompt
> > > The prompt has been added to Appendix I !
> > >
> > > ## Ablation Study
> > > (1) We’ve run HCP+MLP and HCP+Avg for math errors and pairwise comparison. The results are below:
> > >
> > > | LM      | Method                | Math NMSE 100 | Math NMSE 500 | Math CCC 100 | Math CCC 500 | RAG NMSE 100 | RAG NMSE 500 | RAG CCC 100 | RAG CCC 500 |
> > > |---------|-----------------------|---------------|---------------|--------------|--------------|--------------|--------------|-------------|-------------|
> > > | NeoBERT | Gradient Descent      | 1.05          | 1.01          | 0.02         | 0.06         | 1.44         | 1.02         | 0.02        | 0.10        |
> > > | GPT-4.1 | Basic Prompt          | 1.59          | 1.59          | 0.36         | 0.36         | 2.18         | 2.18         | 0.47        | 0.47        |
> > > |         | Detailed Prompt       | 1.13          | 1.10          | 0.39         | 0.41         | 1.94         | 1.92         | 0.49        | 0.50        |
> > > |         | MENTAT (Basic Prompt) | 0.87          | 0.76          | 0.56         | 0.67         | 2.07         | 1.80         | 0.55        | 0.58        |
> > > | GPT-5   | Basic Prompt          | 0.77          | 0.77          | 0.69         | 0.69         | 2.62         | 2.35         | 0.30        | 0.31        |
> > > |         | Detailed Prompt       | 0.78          | 0.78          | 0.69         | 0.69         | 2.18         | 2.01         | 0.33        | 0.33        |
> > > |         | MENTAT (Basic Prompt) | 0.52          | 0.42          | 0.72         | 0.78         | 1.97         | 1.96         | 0.36        | 0.36        |
> > > | GPT-4.1 | MENTAT Prompt         | 1.39          | 1.29          | 0.45         | 0.48         | 2.00         | 1.69         | 0.45        | 0.48        |
> > > |         | MENTAT-Avg            | 1.00          | 1.01          | 0.52         | 0.52         | 1.82         | 1.48         | 0.54        | 0.54        |
> > > |         | Detailed Prompt-Avg   | 0.88          | 0.86          | 0.30         | 0.32         | 1.94         | 1.91         | 0.49        | 0.48        |
> > > |         | Detailed Prompt + MLP | 0.83          | 0.83          | 0.55         | 0.57         | 2.02         | 2.00         | 0.49        | 0.49        |
> > > |         | GEPA                  | 1.04          | 1.07          | 0.45         | 0.47         | 1.91         | 1.89         | 0.50        | 0.49        |
> > > | GPT-5   | MENTAT Prompt         | 0.66          | 0.58          | 0.59         | 0.64         | 1.73         | 1.31         | 0.35        | 0.41        |
> > > |         | MENTAT-Avg            | 0.65          | 0.64          | 0.57         | 0.61         | 1.68         | 1.45         | 0.36        | 0.40        |
> > > |         | Detailed Prompt-Avg   | 0.72          | 0.71          | 0.60         | 0.60         | 1.71         | 1.53         | 0.36        | 0.37        |
> > > |         | Detailed Prompt + MLP | 0.57          | 0.52          | 0.60         | 0.62         | 1.11         | 1.05         | 0.43        | 0.43        |
> > > |         | GEPA                  | 0.78          | 0.63          | 0.68         | 0.69         | 2.48         | 2.29         | 0.28        | 0.28        |
> > >
> > >
> > >
> > > We remark here that this is a *very* expensive experiment to run in full: 3 runs X 3 rollouts x 4 tasks x 2 models (with one of them being GPT5), so we abstain from scaling it up further given that it does *not* pertain to testing any of the claims that the paper intends to make.
> > >
> > > (2) The plot has been added to Figure 5!
> > >
> > > (3) If we understand correctly, this could be an interesting way to improve MENTAT further. Given the cost and complexity of adding new variants of our approach, we have to defer this type of study to future research.

---

> ### Author Response · Authors · 2025-11-27
> **Response to Experiment Scoring**
>
> Thank you for sharing this argument with us. In designing the task, we have indeed had detailed internal discussions on the right way to treat this label from first principles and arrived at the conclusion that including this hard cutoff reduces the validity of the task as a continuous regression problem.
>
> We expect that in a typical application, e.g. during RL for math with verifiable rewards, checking _whether the final answer is correct_ can be done easily with programmatic verifiers. The instances in which a system does make a mistake are kept and, for those, the regression formulation we currently have is the right fit for locating errors.
>
> Additionally, it's unfortunately the case that making this change would further require rerunning the entire Math Errors column, since all learning runs would have to change. This is a very expensive undertaking for a setting that we disagree with in principle (as above) and may not really contribute much new insight, especially given the presence of a few other tasks and the new  inclusion of additional experiments (GEPA and HCP+MLP). Nonetheless, we appreciate the discussion with the reviewer on this topic, and would be happy to elaborate on this reasoning in the paper if desired.

---

### Official Review · Reviewer_pSw3 · 2025-11-04

**Soundness:** 2
**Presentation:** 2
**Contribution:** 2
**Rating:** 4
**Confidence:** 3

**Summary:**

The paper proposes the concept of reasoning-intensive regression, a type of language regression task that requires intensive reasoning from an LLM. To benchmark it, the author designed a benchmark with 4 tasks, spanning different levels of required reasoning capability. Lastly, the author designed a combined evolution and neural aggregation method to better solve this benchmark.

**Strengths:**

The paper is well-written, the benchmark and the methodology are well-presented.

**Weaknesses:**

* While presenting a new benchmark for RIR, the paper lacks an error analysis addressing the current models’ or methods’ pitfalls on the benchmark.
* Additionally, regarding the different levels of regression tasks and their corresponding benchmark designs, it would be better to include some quantitative or qualitative comparisons showing the differences between level-3 reasoning-intensive tasks and levels 2 and 1 (e.g., average CoT length, confidence, self-consistency, etc.).
* For the experiment setup with Mentat, I feel that a very important practical question is how much data is allocated to train or supervise the method (in phase 1 and phase 2). Data efficiency is crucial, especially since collecting regression data is costly. Therefore, a data-point efficiency experiment is needed (beyond the two options of 100 and 500 budget).
* I’m also curious — in Table 1, how would the method compare to few-shot learning with carefully selected few-shot examples? This seems to be an important baseline as well.

**Questions:**

N/A

---

> ### Author Response · Authors · 2025-11-25
> **Qualitative Examples of Failure Cases + Experimentation Addition**
>
> Thank you for taking the time to read this paper, we are grateful for your help, especially related to questions around diving into specific failure cases. We were glad to hear that you found the paper well written with a good presentation of the benchmarks and methodology.
>
> ## Error Analysis
>
> > While presenting a new benchmark for RiR, the paper lacks an error analysis addressing the current models’ or methods’ pitfalls on the benchmark.
>
> We appreciate this feedback! The paper does include analysis of method pitfalls: (1) Figure 1 and Section 3 analyze NeoBERT's distribution collapse and loss "hacking"; (2) Section 4.4 and Figure 5 analyze GPT-5's overthinking and center-seeking behavior on pairwise RAG; (3) Appendix B analyzes systematic quantization bias in LLM numerical outputs. However, we agree that deeper per-task error analysis would strengthen the paper. **We have added, in the appendix, qualitative examples of specific failure cases for each task, and analyze what problem characteristics might correlate with errors (e.g., solution length, problem type for math errors). See Appendix C for this!**
>
> ## Taxonomy
>
> > Additionally, regarding the different levels of regression tasks and their corresponding benchmark designs, it would be better to include some quantitative or qualitative comparisons showing the differences between level-3 reasoning-intensive tasks and levels 2 and 1 (e.g., average CoT length, confidence, self-consistency, etc.).
>
> We agree that explicit comparisons would strengthen the level distinction, although adding something like CoT length, might not be the right direction. Shojaee et al. (2025), which we cite, show models often "overthink" simple tasks, making length an unreliable proxy. Figure 2, outlining the various levels, of which the four datasets we utilize are in level 2 and 3 provides good intuition by what we mean by reasoning intensive regression. **The appendix now includes example entries from each of the datasets, moreover, we plan to release all datasets used in this paper in a supplementary code repository!**
>
> ## Sampling Analysis
>
> > For the experiment setup with Mentat, I feel that a very important practical question is how much data is allocated to train or supervise the method (in phase 1 and phase 2). Data efficiency is crucial, especially since collecting regression data is costly. Therefore, a data-point efficiency experiment is needed (beyond the two options of 100 and 500 budget).
>
> We agree that data efficiency is a crucial aspect of RiR. For this reason, we study two realistic settings where only 100 or 500 datapoints are available for training , and find that our method is very compelling in both cases. Especially given the small total sizes of these datasets (math errors is ~1700, while Pairwise RAG is ~3500, while essay grading is ~3700).  **Is there a specific weakness that you see in this experiment design? If so, is there a specific size or experiment that would strengthen that, keeping in mind that the cost of running each cell sufficiently many times (for averaging) and the limited gain of information from adding more sizes.**
>
> > I’m also curious — in Table 1, how would the method compare to few-shot learning with carefully selected few-shot examples? This seems to be an important baseline as well.
>
> We respectfully argue that few-shot learning introduces key challenges such as example selection strategy, score coverage, and context length. This introduces significant design decisions that would themselves require optimization. MENTAT's prompt evolution can be viewed as learning an implicit representation of good examples through error analysis. **We do agree that we can robustify the baselines (as suggested by another reviewer) and are adding GEPA (a very popular and new prompt optimizer) as another baseline. See an updated Table 1. We will update Table 2 shortly.**
>
>
> **Thank you again for taking the time to review the paper and providing helpful feedback! Do the above actions address your concerns with the paper? If not, what further clarification or modifications could we make to improve it?**

---

### Official Review · Reviewer_MtER · 2025-11-04

**Soundness:** 3
**Presentation:** 4
**Contribution:** 3
**Rating:** 6
**Confidence:** 3

**Summary:**

The paper defines a new task family called Reasoning-Intensive Regression (RiR)—language-based regression problems that require explicit multi-step reasoning instead of shallow feature extraction. The authors argue that standard encoder fine-tuning collapses to narrow score ranges, while directly prompting frozen LLMs yields coarse, discretized outputs.
This paper establish an initial benchmark of four diverse RiR tasks (mathematical-error detection, instruction-following quality, pairwise RAG comparison, and essay grading) and propose MENTAT, a lightweight two-phase method combining batch-reflective prompt optimization with neural ensemble aggregation.
Across all tasks, MENTAT improves Concordance Correlation Coefficient (CCC) over both encoder and prompting baselines, showing that simple prompt evolution plus multi-rollout calibration can recover both reasoning depth and numeric precision.

**Strengths:**

1. The proposed Reasoning-Intensive Regression (RiR) benchmark captures an emerging but under-explored problem space—tasks requiring both deep reasoning and precise numeric scoring (e.g., reward modeling, rubric-based evaluation). This formulation is well-motivated and relevant to ongoing LLM-judging and alignment research.
2. The authors convincingly argue that normalized MSE can be misleading due to output collapse and advocate using the Concordance Correlation Coefficient (CCC), which jointly considers variance, mean alignment, and correlation. This insight is both intuitive and empirically supported, representing a meaningful contribution to LLM evaluation methodology.
3. MENTAT’s two-phase structure—batch-reflective prompt optimization followed by neural aggregation—is simple, effective, and closely mirrors how humans iteratively refine prompts and aggregate judgments. It provides measurable gains while remaining lightweight and reproducible across models.

**Weaknesses:**

1. The mathematical RiR task relies on ProcessBench, which explicitly annotates the first erroneous reasoning step. While this allows continuous regression scoring, such step-level annotations are rare in most math datasets, limiting scalability and generalizability.
2. Instruction-Following uses gpt-oss-20b (an open-source LLM) while other tasks use GPT-5 or GPT-4.1. The authors briefly claim this is “for reproducibility and generalization validation,” but no comparison is given to show GPT-5 behaves similarly on that task.
3. Although the paper claims MENTAT is lightweight, it involves multiple LLM inference rounds and K-fold rollouts per sample. The actual compute cost, token budget, or latency trade-off is not quantified. Moreover, comparisons to recent prompt-optimization or ensemble calibration methods are absent, leaving uncertainty about relative efficiency.

**Questions:**

1. Your prompt evolution stage always selects approximately √n of the worst samples for reflection. Have you experimented with smaller or larger batches, or grouping by error type rather than just score percentile?
2. Could the MLP aggregation in Phase 2 be replaced by a non-trainable closed-form rule (e.g., median plus variance correction)? If possible, please provide a comparison table between simple aggregation baselines (average, median, trimmed mean) and the learned MLP.

---

> ### Author Response · Authors · 2025-11-25
> **Computation Cost + GEPA Additions**
>
> Thank you for taking the time to read this paper, we are grateful for your help, especially related to questions of computation cost. We were glad to hear that you found our problem definition of RiR timely, the introduction of CCC as an important metric for these types of problems, and for the simplicity of MENTAT.
>
> ## Process Bench
>
> > The mathematical RiR task relies on ProcessBench, which explicitly annotates the first erroneous reasoning step. While this allows continuous regression scoring, such step-level annotations are rare in most math datasets, limiting scalability and generalizability.
>
> This is a valid point about annotation requirements, however, this comment supports rather than undermines our contribution: RiR benchmarks are scarce precisely because such annotations are difficult to obtain. Our goal is to establish an initial benchmark framework and demonstrate that RiR poses distinct challenges, not to claim unlimited scalability. We hope to see more RiR style benchmarks in the future!
>
> ## Instruction Following and OSS
>
> > Instruction-Following uses gpt-oss-20b (an open-source LLM) while other tasks use GPT-5 or GPT-4.1.
>
> To understand this experimental design, it may be helpful to learn that we started by designing three tasks and tested two (closed) OpenAI models. Subsequently, we sought to simultaneously _expand_ our study to open-weights models and to include a fourth task (Instruction-Following). Instruction-Following originally uses fine-grained labelings from Llama3.1-70B (as described in Section 2), but in doing so we do not expect that, even in principle, it would have room to test a superior model as capable as GPT-5. Thus, the Instruction-Following task seeks mainly to demonstrate that RiR patterns appear beyond large and proprietary models.
>
> ## Computation Cost
>
> > Although the paper claims MENTAT is lightweight, it involves multiple LLM inference rounds and K-fold rollouts per sample. The actual compute cost, token budget, or latency trade-off is not quantified. Moreover, comparisons to recent prompt-optimization or ensemble calibration methods are absent, leaving uncertainty about relative efficiency
>
> This is a great suggestion, thank you so much! **Results from the optimizer, GEPA, are in the updated Table 1, we will update Table 2 shortly. We’ve also added a section 4.3.1 talking about the computation cost differences between MENTAT and GEPA.** Essentially, during optimization (Phase 1), MENTAT uses a fixed $3$-iteration design for training sets of $n=250$ samples. Each iteration evaluates the current prompt on all $250$ samples (parallelizable) and performs one reflection call analyzing the $\sqrt{250} \approx 16$ worst-performing examples, yielding approximately $753$ total LLM  calls across 3 sequential steps.
>
> We compare against GEPA, a recent prompt optimizer. Using GEPA's “light" configuration on $n=250$, we observe convergence after an average of $23$ sequential iterations (ranging $15-34$ across runs). While both methods can parallelize evaluation within iterations, GEPA's evolutionary search requires substantially more sequential rounds than MENTAT's fixed 3-iteration design. In deployment scenarios where wall-clock time is critical, MENTAT's architectural advantage of 3 versus approximately 23 sequential bottlenecks provides meaningfully faster optimization, though we note that total token consumption may differ based on GEPA's candidate generation strategy.
>
> ## Response to Questions:
>
> > Your prompt evolution stage always selects approximately √n of the worst samples for reflection. Have you experimented with smaller or larger batches, or grouping by error type rather than just score percentile?
>
> We experimented with batch sizes but noticed that smaller batches led to prompt oscillation (fixing one error type while breaking another), while larger batches overwhelmed the reflection model with too many diverse errors to synthesize coherently. Grouping by error type is an interesting direction but requires either some pre-defined error taxonomies per task, or an additional clustering step, which would both add complexity. **We leave error-type-aware batch selection to future work and will note this in the paper.**
>
> > Could the MLP aggregation in Phase 2 be replaced by a non-trainable closed-form rule (e.g., median plus variance correction)? If possible, please provide a comparison table between simple aggregation baselines (average, median, trimmed mean) and the learned MLP.
>
> We remark here that Table 1 already includes this comparison: MENTAT-Avg uses averaging instead of the learned MLP after getting the optimized prompt from Phase I. Results show the MLP provides consistent (though sometimes modest) gains over averaging!
>
>
>
> **Thank you again for taking the time to review the paper and providing helpful feedback! Do the above actions address your concerns with the paper? If not, what further clarification or modifications could we make to improve it?**

---

> > ### Comment · Reviewer_MtER · 2025-11-27
> >
> > Thank you for the detailed and thoughtful response. The clarifications and the additional comparisons are helpful.
> > I appreciate the effort put into addressing the concerns I raised.

---

### Author Response · Authors · 2025-12-03
**General Response**

We thank the reviewers for their suggestions, which have helped improve the presentation, evaluation, and analysis in our paper. We are also encouraged that the reviewers appreciated: (1) the definition of the new reasoning-intensive regression (RiR) problem and the construction of taxonomy levels, (2) the emphasis on the importance of a different metric, CCC, for properly reporting performance on RiR tasks, (3) the collection of four tasks (math error detection, instruction following, pairwise RAG comparison, and essay grading) for RiR, (4) the development of MENTAT, a method combining batch-reflective prompt optimization with neural aggregation for RiR tasks, and (5) our robust ablation studies comparing against baseline methods (fine-tuning NeoBERT, prompting advanced LLMs).

With the reviewers' suggestions, we were able to make several improvements throughout the paper. The highlights include: the performance of GEPA, a state-of-the-art optimizer, added to our evaluation and compared against MENTAT and MENTAT-Prompt (see updated Table 1); an analysis of the variance distribution of MENTAT (lines 286-308); further details on the computational cost of MENTAT (Section 4.3.1); an analysis of the failure modes of our RiR tasks (Appendix C); and additional example entries from our RiR datasets along with snippets of the error analysis and prompt refinement code (Appendices E & I).

Some reviewers questioned specific task design decisions. For Mathematical Error Detection, one reviewer advocated including correct solutions from ProcessBench. We maintain that assigning any scores to correct solutions (neither 0.0 nor 10.0 is semantically appropriate for "no error") would transform continuous regression into classification with arbitrary labels. In applications like RL for math, programmatic verifiers check answer correctness; the regression challenge lies in localizing errors within incorrect solutions. Similarly, our adaptations for Instruction Following (continuous scores with harmonic mean) and Pairwise RAG (raw judgments vs. Arena rankings) deliberately preserve continuous regression structure rather than collapsing to ordinal or near-binary outcomes. Finally, one reviewer noted that including NeoBERT (a finetuned encoder) alongside frozen LLMs crosses model regimes, but this is an important experimental decision  validating that RiR poses challenges across standard approaches, motivating methods like MENTAT that bridge these paradigms.

We address specific questions and detailed feedback in our individual responses to each reviewer in the individual responses.

---

### Meta-Review · Area_Chair_R1cX · 2026-01-02

**Summary:**

The paper identifies and formalizes Reasoning-Intensive Regression (RiR), a class of tasks where models must produce precise numerical scores based on complex textual reasoning. The authors argue that traditional metrics like MSE are insufficient for these tasks due to variance collapse and advocate for the Concordance Correlation Coefficient (CCC). They contribute a four-task benchmark and MENTAT, a lightweight framework combining batch-reflective prompt optimization with neural ensemble aggregation.

The reviewers generally agreed on the task definition and the insightful use of CCC. However, concerns were raised regarding the formalism of the RiR taxonomy, the novelty of MENTAT compared to existing prompt optimizers, and the specific casting of tasks (notably the math error detection task).

**Reviewer Concerns:**

Addressed concerns:
- The authors included GEPA (a modern prompt optimizer) as a baseline, demonstrating that MENTAT provides better parallelism and higher CCC. They also quantified the computational trade-offs, answering questions from reviewer MtER
- The authors provided a new table during the rebuttal comparing Detailed Human-Crafted Prompts (HCP) + MLP/Avg against MENTAT to answer reviewer yiZN's question
- Authors added rollout variance distributions showing that MENTAT prompts reduce internal model inconsistency

Outstanding concerns:
- The three-level taxonomy remains largely heuristic. While the authors provided empirical evidence (NeoBERT failing on Level 3 but succeeding on Level 2), a rigorous, first-principles definition of what constitutes Reasoning-Intensive vs. Shallow is still absent, and not solved regarding reviewer pSw3 and 2fyW's concerns
- Reviewer yiZN strongly insisted on including correct solutions in the math benchmark. The authors' refusal (arguing it shifts regression to classification) remains a point of structural disagreement that affects the benchmark's perceived completeness
- While the authors clarified how MENTAT differs from TextGrad or APO by reflecting on large batches simultaneously, some reviewers still view the contribution as an incremental combination of existing prompt engineering and self-consistency techniques.

**Reviewer Scores:**

Based on the outstanding concerns raised by multiple reviewers, reviewer 2fyW and WmpZ tend to maintain the reject score unchanged because the initial poor presentation rating is hard to fully overcome and the disagreement with task casting and mixing model regimes

---

### Decision · Program_Chairs · 2026-01-26

Reject